# VaporTok: RL-Driven Adaptive Video Tokenizer with Prior & Task Awareness

**Minghao Yang**[1][*]   **Zechen Bai**[2][*]   **Jing Lin**[3]   **Haoqian Wang**[1][†]   **Alex Jinpeng Wang**[4][†]

[1]Tsinghua University   [2]National University of Singapore
[3]Nanyang Technological University   [4]Central South University

## Abstract

Recent advances in visual tokenizers have demonstrated their effectiveness for multimodal large language models and autoregressive generative models. However, most existing visual tokenizers rely on a fixed downsampling rate at a given visual resolution, and consequently produce a constant number of visual tokens, ignoring the fact that visual information of varying complexity warrant different token budgets. Motivated by this observation, we propose an adaptive video tokenizer "VaporTok" with two core contributions: **Probabilistic Taildrop**: We introduce a novel taildrop mechanism that learns a truncation index sampling distribution conditioned on visual complexity of the video. During both training and inference, the decoder reconstructs videos at adaptive token lengths, allocating more tokens to complex videos and fewer to simpler ones. **Parallel Sample GRPO with Vapor Reward**: By leveraging the probability distribution produced by probabilistic taildrop, we reformulate the visual tokenization pipeline as a sequential decision process. To optimize this process, we propose a variant of GRPO and a composite reward encompassing token efficiency, reconstruction fidelity, and generative quality, thus enabling metrics-aware adaptive tokenization across diverse objectives. Extensive experiments on standard video generation benchmarks confirm our analysis, showing that our adaptive approach matches or outperforms fixed-rate baselines and naive taildrop while using fewer tokens.

## 1  Introduction

Visual generative models have undergone rapid advancements in recent years, progressing from VAEs[22] and GANs[17] to diffusion models[18]. More recently, autoregressive (AR) based approaches[7, 5, 42, 45] have emerged as a prominent direction, demonstrating competitiveness with diffusion models. The superior performance is largely due to the scalability and flexibility of the AR paradigm, as demonstrated by large language models (LLMs) [58, 46, 10]. Similar to LLMs, AR-based visual generative models necessitate a visual tokenizer, which is essential for converting image or video data into vector representations that the model can process. Consequently, research into visual tokenizers has become a central focus in visual generative models.

Despite their remarkable performance, AR models still face fundamental limitations inherent in the AR paradigm. Specifically, the computational complexity of processing token sequences grows quadratically with their length. In addition, the prediction errors can accumulate progressively as observed in numerous works[1, 8, 36]. Intuitively, a shorter, more compact visual token sequence can be a favorable option. Existing visual tokenizers[47, 60, 29, 62] generally output a predetermined number of latent tokens for subsequent generation tasks. While there are some attempts at adaptive

---

[*]Equal contribution.
[†]Corresponding authors.

39th Conference on Neural Information Processing Systems (NeurIPS 2025).

tokenization (e.g., [11, 57, 2, 31, 51]), most of them still depend on a manually specified range of token counts without an effective prior, limiting their capacity for truly adaptive tokenization.

Another representative limitation in AR-based visual generative models is that the training of tokenizers and AR models are usually divided into two separated stages, making the visual tokenizer sub-optimal and preventing it from generalizing well to downstream tasks. In fact, the above issues are intertwined. Determining visual representation and its adaptivity should not only satisfy the data prior but also align with downstream task characteristics. On the other hand, the supervision of visual generation can help optimize both AR generative model and, more importantly, its tokenizer, *if*

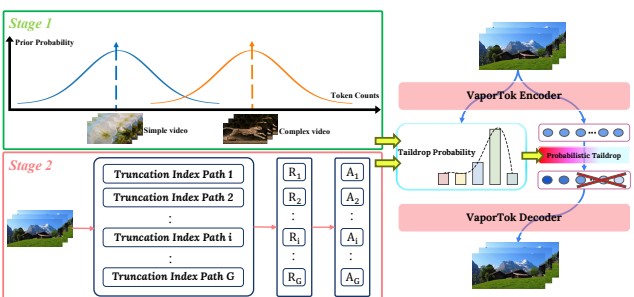

Figure 1: **VaporTok employs two-stage training:** the first stage uses visual complexity for supervision, the second stage uses GRPO with multiple task-aware rewards for supervision.

*back-propagated properly*. This implies that an efficient, adaptive and downstream-aware visual tokenizer is the key to address the AR issues mentioned above.

In this work, we introduce VaporTok, an adaptive video tokenizer that leverages both data prior and task-related signals to optimize its adaptivity. Specifically, we observe that **visual data inherently possesses varying degrees of complexity in terms of content across spatial and temporal dimensions.** Consequently, representing simple content with excessive tokens can lead to redundancy. Conversely, complex content may not be adequately captured if represented with too few tokens. Therefore, dynamically adjusting token number based on visual complexity would enable a more faithful alignment between visual information and its representation. Inspired by this, we propose "Probabilistic Taildrop", a method that leverages visual complexity to build a sampling distribution over token counts and then drops the excess tail tokens accordingly. The complexity informed taildrop not only helps mitigate quadratic computation by producing a compact token sequence, but also reduce error accumulation by condensing meaningful information at the head of the sequence.

When tackling the limitation of training disparity between tokenizer and AR model, the main technical challenge is the differentiability of the two models due to hard token indexing. Recently, reinforcement learning (RL), particularly GRPO[37], has shown considerable advantages in various domains[10, 13, 6, 59, 50, 21]. Notably, **the reward formulation in reinforcement learning does not necessitate differentiability with respect to the parameters of the policy model**. Therefore, we propose to leverage this characteristic to unlock task-aware adaptive tokenizer training. The core idea is that by optimizing the visual tokenizer using RL-based rewards, these non-differentiable supervisory signals could be effectively transmitted during its training. Although the idea seems intuitive, it is non-trivial to achieve. Our approach innovatively formulates visual tokenization as a sequential decision process, which is compatible with RL training framework. **To the best of our knowledge, this is the first work that employs RL framework to formulate and train a visual tokenizer for task-aware adaptive tokenization.** In addition, we design a novel "Vapor Reward" that accommodates multiple supervisory signals into the reward function, providing valuable insights for the community. Our contribution can be summarized as follows:

- **Probabilistic Taildrop**: An adaptive video tokenization technique adjusting token count based on visual complexity for more efficient sequences.

- **RL for Task-Aware Tokenizer Training**: A novel framework that formulates video tokenization as a sequential decision process and uses RL to optimize the tokenizer, making its adaptivity aware of multiple metrics, including downstream AR generation performance.

- **Vapor Reward**: A new multi-signal reward function designed to effectively guide the RL-based tokenizer optimization.

## 2 Related work

**Adaptive visual tokenizers.** Most existing visual tokenizers [12, 29, 62, 27, 63, 48, 33, 9, 28, 49, 16, 68, 20, 55, 43, 67] are limited to representing visual features using a fixed number of tokens, ignoring the varying complexity of visual content. As a result, recent works have shifted toward adaptive tokenization schemes that dynamically adjust the token budgets based on visual complexity. ALIT [11] employs a recurrent encoding scheme to progressively assemble the token sequence of an image and the process can be halted. CAT[39] leverages a MLLM's complexity analysis of image captions to regress which compression rate to apply. FlexTok[2] and One-D-Piece[31] both employ nested dropout to encourage the model to prioritize core visual information in early tokens, yielding a coarse-to-fine representation without fixed-length constraints. ElasticTok[57] extends the taildrop technique to the video domain and adaptively selecting the token count at inference based on a reconstruction-quality threshold. ViLex[51] introduces a novel "visual language" that encodes image tokens after taildrop into the textual token space by self-supervised training on a frozen text-to-image diffusion model. These adaptive methods either pick from a fixed set of token counts or use random taildrop with thresholding, yet neither is truly adaptive: the first is limited to predefined choices, and the second ignores visual complexity during training. In addition, they focus only on reconstruction and neglect the impact of adaptive token selection on downstream performance.

**GRPO in vision domains.** Enhancing foundation models via reinforcement learning has become a major research focus. Motivated by the strong inference performance of DeepSeek R1[10], Group Relative Policy Optimization (GRPO) [37] has demonstrated clear advantages over PPO in both training efficiency and final model quality. In computer vision, core generation and understanding tasks, including Visual Question Answering[32, 59], Image Grounding[6], Video Question Answering[14, 38], and Visual Generation[50, 56, 21, 25], are actively investigating the integration of GRPO to boost existing methods, seeking to transfer the success of GRPO from large language models to vision.

## 3 Method

### 3.1 Probabilistic Taildrop

Conventional taildrop technique [23] simply samples truncation positions from *uniform distribution* without considering any prior information about visual complexity. This can lead to insufficient preservation of complex visual content when too few tokens are selected, and to unnecessary redundancy when too many tokens are retained for simpler visuals. In contrast, we propose probabilistic taildrop, which constructs a truncation-sampling distribution informed by the complexity of the visual input. During training, truncation index are drawn from this distribution to perform taildrop. This strategy **both preserves the fundamental principle of taildrop—compressing semantic information into earlier tokens while relegating detailed information to later tokens—and adaptively selects truncation points based on visual priors, thereby achieving efficient yet faithful visual encoding**.

To implement probabilistic taildrop, we introduce the Taildrop Probability Query Module in Section 3.1.1 to obtain the taildrop probabilities. To incorporate visual priors into the supervision of these probabilities, we construct a distribution from the visual information to regularize the predicted taildrop probabilities, as described in Section 3.1.2.

### 3.1.1 Taildrop Probability Query Module

Given a video $V \in \mathbb{R}^{T \times H \times W \times 3}$, VaporTok first patchify it into a sequence of video tokens $P$ :

$$P = \text{Patchify}(V) \in \mathbb{R}^{(\frac{T}{f_T} \times \frac{H}{f_H} \times \frac{W}{f_W}) \times D}, \tag{1}$$

where $f_T, f_H, f_W$ are the temporal and spatial downsampling factors. And then $P$ will be concatenated with $K$ learnable query tokens $Q \in \mathbb{R}^{K \times D}$ and the combined sequence will be passed into the encoder :

$$Z_P \oplus Z_Q = \text{Enc}\big(P \oplus Q\big) \in \mathbb{R}^{(\frac{T}{f_T} \times \frac{H}{f_H} \times \frac{W}{f_W} + K) \times D}, \tag{2}$$

where $\oplus$ denotes concatenation and $Z_P, Z_Q$ denotes the represatation of $P, Q$ after encoder respectively. To enable VaporTok to learn a distribution for taildrop, we introduce Taildrop Probability

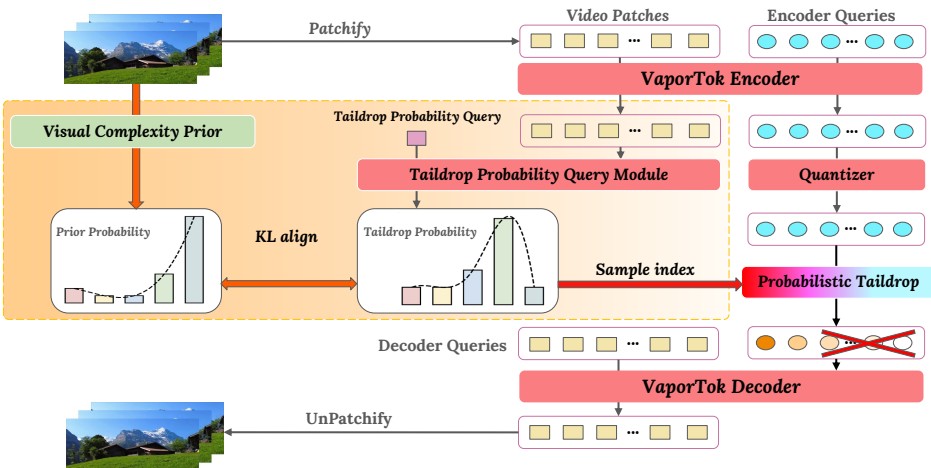

Figure 2: **VaporTok pipeline:** A taildrop probability query module constructs a taildrop probability supervised by a video-complexity prior. An index is then sampled according to this probability, and only the tokens preceding that index are retained for reconstruction training.

Query Module consisting of $I$ successive transformer blocks and a softmax layer, as illustrated in Figure 2. A dedicated taildrop probability query $Q_{\text{tail}} \in \mathbb{R}^D$ is concatenated with $Z_P$ and fed into transformer blocks of Taildrop Probability Query Module :

$$Q'_{\text{tail}} \oplus Z'_P = \text{TransformerBlocks}\big(Q_{\text{tail}} \oplus Z_P\big) \in \mathbb{R}^{(1 + \frac{T}{f_T} \times \frac{H}{f_H} \times \frac{W}{f_W}) \times D}, \tag{3}$$

$Q'_{\text{tail}}$ is then passed through an MLP followed by a softmax layer to produce the taildrop probability distribution $P$ :

$$P = \text{Softmax}\Big(\text{MLP}\big(Q'_{\text{tail}}\big)\Big) \in \mathbb{R}^K, \tag{4}$$

where $K$ is the total token counts of latent query $Z_Q$. To enable adaptive token usage, a truncation index $t$ is sampled according to the learned taildrop probability $P$ :

$$t \sim \text{Categorical}(P) \tag{5}$$

All latent tokens of $Z_Q$ whose index exceeds the sampled index $t$ are discarded and only the first $t$ tokens are concatenated with the decoder query $M$ which are then passed to the decoder for reconstruction :

$$PTD\_Z_Q = \text{ProbabilisticTaildrop}(Z_Q) = Z_{Q,\,1:t} \in \mathbb{R}^{t \times D} \tag{6}$$

$$\hat{V} = \text{Dec}\,(M \oplus PTD\_Z_Q) \in \mathbb{R}^{T \times H \times W \times 3} \tag{7}$$

The first-stage training loss of VaporTok is composed of L1 reconstruction loss, LPIPS perceptual loss[66], GAN loss[17], quantizer loss[48] and prior loss[48]. The detail is provided in the Appendix.

### 3.1.2 Video Complexity Prior

Implicitly modeling video complexity is impractical. Therefore, we explicitly supervise the Taildrop Probability Query Module with a Gaussian distribution that contains visual prior information. First, the spatial and temporal complexities are computed separately and then these complexities are mapped to a corresponding token count via Equation 9. We then construct a Gaussian distribution to supervise the taildrop probabilities, with its mean set to the token count obtained from the preceding mapping and its variance determined by $K$, the total number of encoder query tokens. Specifically, let visual complexity of video $i$ be :

$$c_i = \text{SC}_i \times \text{TC}_i, \tag{8}$$

where $\text{SC}_i$ and $\text{TC}_i$ are its spatial and temporal complexities whose details are provided in the Appendix. Then maintain an empirical CDF $F$ over all observed complexities in training set and map each video complexity $c_i$ to a token count $k_i$ by :

$$k_i = \lfloor s + \Delta\,F(c_i) \rfloor, \quad F(c_i) \in [0, 1]. \tag{9}$$

where $s$ is a predefined minimum token count tolerable for reconstruction and $\Delta$ is the maximum increment. This yields a whole dataset wise mapping :

$$\{c_i\}_{i=1}^{|dataset|} \longmapsto \{k_i\}_{i=1}^{|dataset|}. \tag{10}$$

For each training sample $i$, we construct a 1D Gaussian distribution with mean $k_i$ and variance $K \cdot \sigma_{\text{scale}}$ as the prior distribution :

$$GaussianPrior_i = \mathcal{N}\left(t;\, \mu = k_i,\, \sigma^2 = K \cdot \sigma_{\text{scale}}\right) \tag{11}$$

where $k_i$ is the token count calculated by visual complexity prior for the $i$-th video, $K$ is the total token counts of $Z_Q$, and $\sigma_{\text{scale}}$ is a hyperparameter to control the trend of the prior distribution.

Then, the loss of sample video $i$ to train the taildrop branch, specifically Taildrop Probability Query Module and VaporTok encoder, is calculated as the KL divergence between $P_i$ and $GaussianPrior_i$:

$$Loss_i = \text{KL}(P_i \,\|\, GaussianPrior_i) \tag{12}$$

## 3.2 Parallel Sample GRPO with Vapor Reward

Since the sampling operation employed in the VaporTok is non-differentiable, it is infeasible to propagate the reconstruction loss to the Probabilistic Taildrop branch through the latent space. Nevertheless, in reinforcement learning, the reward function can be treated as an arbitrary black box, whose information can be passed to the policy model, even though the reward is not differentiable with respect to the parameters of the policy model. In addition, except for the basic reconstruction feedback, **several helpful metrics can also be incorporated in reward definition to make the tokenizer more efficient and appropriate for downstream generation task.**

To this end, we cast the video tokenization as a sequential decision process in Section 3.2.1 and introduce Parallel Sampling GRPO in Section 3.2.2 to optimize this process while avoiding mode collapse. Furthermore, in Section 3.2.3, we introduce Vapor Reward, which enables GRPO to refine VaporTok's adaptivity with respect to both reconstruction and generation tasks. Finally, in Section 3.2.4, we define the optimization objective of Parallel Sample GRPO.

### 3.2.1 Definition of sequential decision process

Given an entire video $V_{entire} \in \mathbb{R}^{N_{\text{GRPO}} \times H \times W \times 3}$ of length $N_{\text{GRPO}}$, we first partition it into $L = \left\lceil \frac{N_{\text{GRPO}}}{N_{\text{VAE}}} \right\rceil$ video clips $(V_{clip}^1, V_{clip}^2, \ldots, V_{clip}^i, \ldots, V_{clip}^L)$, where $N_{\text{VAE}}$ is the predefined frame count can be processed by VaporTok encoder at one time. For each video clip $V_{clip}^i$, VaporTok encoder computes its latent representation $Z_Q^{(i)}$ and Taildrop Probability Query Module computes its taildrop probability distribution $P_i$. We define the three key components of the sequential decision process in the context of token truncation problem as follows:

- **State** $\mathcal{S}$: the current input video clip $V_{clip}^i \in \mathbb{R}^{N_{\text{VAE}} \times H \times W \times 3}$ ;

- **Action** $\mathcal{A}$: the truncation index sampled from $P_i$ to truncate the latent representation $Z_Q^{(i)}$ of current video clip $V_{clip}^i$;

- **Reward** $\mathcal{R}(s, a)$: we will introduce our proposed Vapor Reward in Section 3.2.3.

Then we can get a taildrop probability sequence $(P_1, P_2, \ldots, P_i, \ldots, P_L)$ to sample truncation index for each latent representation $Z_Q^{(i)}$ respectively. Also, we define the policy in our pipeline as $\pi_\theta(P_i \mid V_{clip}^i)$, where $\theta$ is the parameter of Taildrop Probability Query Module proposed in Section 3.1.1. Notably, the key difference between Markov Decision Process (MDP) and our proposed sequential decision process is that the latter does not satisfy the Markov property, and its transition dynamics $\mathcal{P}(s' \mid s, a)$ are implicitly defined by the entire input video $V_{entire}$.

### 3.2.2 Parallel Sample GRPO

In our setting, the $G$ trajectories within GRPO group share exactly the same state sequence $(V_{clip}^1, V_{clip}^2, \ldots, V_{clip}^i, \ldots, V_{clip}^L)$, and thus the same policy $\pi_\theta(P_i \mid V_{clip}^i)$ is applied. This leads to

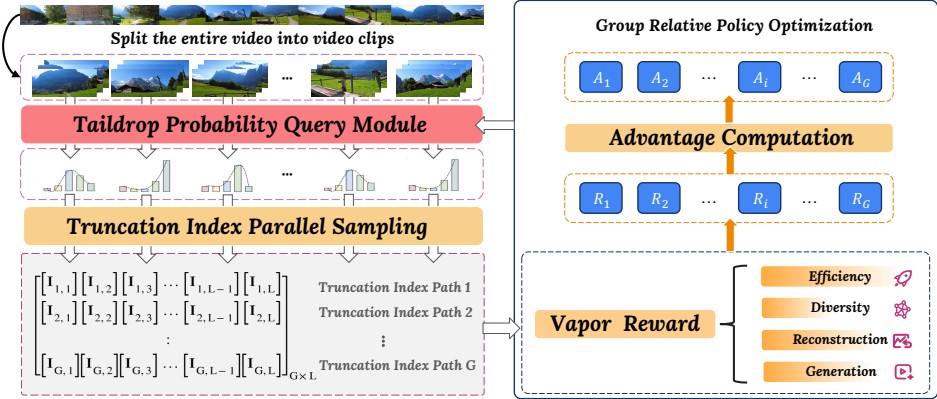

Figure 3: **Parallel Sample GRPO pipeline:** Split the video into a sequence of clips and apply the taildrop probability query module to generate corresponding taildrop probabilities. Then perform parallel sampling to derive a truncation matrix $I$, compute rewards using the Vapor Reward, and update the parameters of taildrop probability query module via GRPO.

highly similar or even identical sampled truncation index path across all $G$ trajectories within a group, especially under greedy or top-k sampling strategies, which in turn causes severe mode collapse. To address this issue, we propose Parallel Sample GRPO, a variant of GRPO specifically designed for training GRPO in such non-Markovian scenario, which concludes two remedies:

**Stochastic parallel sampling.** Rather than restricting sampling to the top-k candidates, we draw samples from the whole categorical distribution defined by the taildrop probability distribution $P_i$. The sampling strategy increases the probability of selecting lower probability indexes and allows for greater diversity between trajectories within a group. Specifically, a truncation matrix $\mathbf{I} = [I_{i,j}] \in \mathbb{Z}^{G \times L}$ is obtained, where each row represents a truncation path that specifies how many tokens are used to represent each clip in the video and there are $G$ such truncation paths in total. Besides, to further mitigate the mode collapse introduced by the non-Markovian nature of the defination, we augment the reward signal with an exploration bonus defined in Equation 14, which encourages diversity across trajectories and therefore promotes exploration.

### 3.2.3 Vapor Reward

We define the Vapor Reward to incorporate reconstruction, token-count, and downstream feedback into the policy model (Taildrop Probability Query Module) while avoiding collapsed sampling paths within a group. The Vapor Reward comprises of following four types of rewards:

**Efficiency reward.** To encourage the latent representation of entire video $V_{entire}$ to be more efficient, we define the efficiency reward. Specifically, for the $i$-th path, the efficiency reward is defined as:

$$R_{\text{efficiency}}^{(i)} = N_{max} - N_{\text{curr}}^{(i)} \tag{13}$$

where $N_{\max}$ is the maximum permissible token count for the video, which can be calculated as $N_{max} = L \times K$, where $K$ is the total token number of $Z_Q$. $N_{\text{curr}}^{(i)}$ is the actual number of tokens retained and can be calculated as $N_{\text{curr}}^{(i)} = \sum_{j=1}^{L} I_{i,j}$

**Diversity reward.** To encourage exploration and mitigate mode collapse, we define a diversity reward for each path based on how different its sampled index sequence is from others in the same group. Specifically, for the $i$-th path, the diversity reward is defined as:

$$R_{diversity}^{(i)} = \frac{1}{(G-1)L} \sum_{j=1}^{L} \sum_{\substack{k=1 \\ k \neq i}}^{G} \mathbf{1} \left[ I_{i,j} \neq I_{k,j} \right] \tag{14}$$

where $\mathbf{I} = [I_{i,j}] \in \mathbb{Z}^{G \times L}$ is the sampled truncation matrix, $G$ is the number of sampled paths, and $L$ is the sequence length. The indicator function $\mathbf{1}[\cdot]$ is equal to 1 if its argument is true and 0 otherwise. A higher reward is assigned to a path if its sampled indices are more dissimilar from the others.

**Reconstruction reward.** To encourage the refined taildrop probability to preserve the reconstruction ability learned from former stage, we define reconstruction reward. Specifically, for the $i$-th path, the reconstruction reward is defined as:

$$R_{\text{reconstruction}}^{(i)} = -\sum_{j=1}^{L} \text{MSE}\big(\hat{V}_{clip}^{j}, V_{clip}^{j}\big) \tag{15}$$

For $j$-th video clip, we use the truncation index path $I_{i,j}$ to truncate the latent representation and reconstruct the video clip via the VaporTok decoder. Then we compute the mean-squared error between reconstructed video clip $\hat{V}_{clip}^{j}$ and ground-truth video clip $V_{clip}^{j}$. The reconstruction reward for the $i$-th path is the negative of the sum of the reconstruction MSE of all video clips of current entire video.

**Generation reward.** To make the taildrop probability be aware of downstream generation performance, we define generation reward. The former work LARP[48] impose a lightweight AR prior model to encourage the latent space to be more suitable for downstream AR-based generation. Hence, the prior model in LARP[48] is reused in our VaporTok (the detail about lightweight ar prior used in VaporTok is provided in the Appendix), and its top-5 accuracy on latent-token predictions is employed as the AR generation reward, guiding the taildrop probability query module to improve efficiency without sacrificing downstream generation performance. Specifically, for the $i$-th path, the generation reward is defined as:

$$R_{\text{generation}}^{(i)} = \sum_{j=1}^{L} \text{Accuracy}_{\text{top5}}\big(PTD\_Z_Q^j\big) \tag{16}$$

### 3.2.4 Objective of Parallel Sample GRPO

For each entire input video $V_{entire}$, a batch of $G$ candidate truncate index sequence $\{\mathbf{k}_i\}_{i=1}^{G}$, where $\mathbf{k}_i = (I_{i,1}, I_{i,2}, \ldots, I_{i,j}, \ldots, I_{i,L})$, is sampled from the old policy $\pi_{\theta_{\text{old}}}$ and scores by reward:

$$R_i = \sum_{m \in \mathcal{M}} \lambda_m R_m^{(i)} \tag{17}$$

where $m$ is one element of $\mathcal{M} = \{\text{efficiency, diversity, reconstruction, generation}\}$ and the non-negative weights $\{\lambda_m\}$ control the relative importance of each reward component. To obtain relative advantages, the rewards $\{R_i\}$ are normalized by their mean and standard deviation:

$$A_i = \frac{R_i - \text{mean}\{R_1, R_2, \ldots, R_G\}}{\text{std}\{R_1, R_2, \ldots, R_G\}}. \tag{18}$$

The parameter $\theta$ is then updated to maximize the following objective:

$$\mathcal{J}_{\text{GRPO}}(\theta) = \mathbb{E}_{\mathbf{I} \sim \pi_{\theta_{\text{old}}}}\left[\frac{1}{G}\sum_{i=1}^{G}\min\Big(\rho_i A_i,\ \text{clip}(\rho_i, 1-\epsilon, 1+\epsilon)A_i\Big) - \beta D_{\text{KL}}\big(\pi_\theta \,\|\, \pi_{\text{ref}}\big)\right], \tag{19}$$

$$\rho_i = \frac{\pi_\theta(\mathbf{k}_i \mid V_{entire})}{\pi_{\theta_{\text{old}}}(\mathbf{k}_i \mid V_{entire})},$$

$\epsilon$ denotes the clipping threshold, $\beta$ scales the KL-divergence penalty, $\mathbf{I} = [I_{i,j}] \in \mathbb{Z}^{G \times L}$ is the sampled truncation matrix, and $A_i$ represents the advantage estimate for the $i$th sample.

## 4 Experiments

**Dataset.** We conduct video reconstruction and generation experiments using the Kinetics-600[4] and UCF-101[41] datasets. In the first stage, we use $N_{\text{VAE}} = 16$ frame video clips at a spatial resolution of $128 \times 128$ for VaporTok training and evaluation following [48]. In the second stage, Parallel Sample GRPO training operates on full $N_{\text{GRPO}} = 80$ frame videos and the sequence length optimized by GRPO is $L = \left\lceil \frac{N_{\text{GRPO}}}{N_{\text{VAE}}} \right\rceil = \left\lceil \frac{80}{16} \right\rceil = 5$.

**Implementation details.** VaporTok first patchifies the input video into a sequence of tokens. In all experiments, we set the patch sizes to $f_T = 4, f_H = 8, f_W = 8$, so that a $16 \times 128 \times 128$ video clip

Table 1: Comparison of generative video models. VaporTok refers to the evaluation results after Stage 1 training, while VaporTok-GRPO refers to the evaluation results after Stage 2 training. The reported token counts are the average number of tokens used per video.

| Method | #Params | | #Tokens | rFVD↓ | gFVD↓ | |
|---|---|---|---|---|---|---|
| | Tokenizer | Generator | | | K600 | UCF |
| *Diffusion-based generative models with continuous tokenizers* | | | | | | |
| VideoFusion [26] | — | 2B | — | — | — | 173 |
| HPDM [40] | — | 725M | — | — | — | 66 |
| *MLM generative models with discrete tokenizers* | | | | | | |
| MAGVIT-MLM [61] | 158M | 306M | 1024 | 25 | 9.9 | 76 |
| MAGVIT-v2-MLM [62] | — | 307M | 1280 | **8.6** | **4.3** | 58 |
| *AR generative models with discrete tokenizers* | | | | | | |
| CogVideo [19] | — | 9.4B | 2065 | — | 109.2 | 626 |
| TATS [15] | 32M | 321M | 1024 | 162 | — | 332 |
| MAGVIT-AR [61] | 158M | 306M | 1024 | 25 | — | 265 |
| MAGVIT-v2-AR [62] | — | 840M | 1280 | **8.6** | — | 109 |
| OmniTokenizer [49] | 82.2M | 650M | 1280 | 42 | 32.9 | 191 |
| LARP-1024 [48] | 173M | 632M | 1024 | 20 | 5.1 | **57** |
| LARP-512 [48] | 173M | 632M | 512 | 53.3 | — | 86 |
| VaporTok (Ours) | 195M | 632M | **498** | 53.9 | 8.3 | 80 |
| VaporTok-GRPO (Ours) | 195M | 632M | **361** | 66.6 | 10.4 | 98 |

is split into $4 \times 16 \times 16 = 1024$ patches. The number of encoder query tokens is set to $k = 1024$. The quantizer and prior model is set as same as [48], where the factorized codebook is employed of size 8192 with embedding dimension $d_{codebook} = 8$ and prior model is adapted from a small GPT-2 backbone[35]. For taildrop probability query module, we set the number of transformer blocks as $I = 2$, and the softmax temperature is set to 1.8. Due to the high computational cost of training, we trained for 30 epochs on the UCF101 and K600 datasets using the pretrained model provided by LARP[48], which required 90 hours on 8 A100 GPUs.

For parallel sample GRPO, we set the group size $G = 8$, the KL penalty weight $\beta = 0.1$, the number of inner iterations $\mu = 2$, and the clipping bounds to $\epsilon_{\text{low}} = 0.2$ and $\epsilon_{\text{high}} = 0.28$ as in [64]. The default reward weights for efficiency, penalty, diversity, reconstruction, and generation are set to 1:1:1:1:1. The GRPO training process uses the UCF101 dataset for a single epoch, which takes 3 hours on a single A100 GPU.

For AR generative model, we adopt a LLaMA-style transformer [42]. In the class-conditional generation task on UCF-101 we prepend a `[cls]` token to represent the category, and a `[stop]` token to cease the generation process when encountering it. The generation task is trained on the UCF101 dataset for 3000 epochs, which takes 40 hours on 8 A100 GPUs.

Table 2: Comparison of different training techniques.

| Base Model | #Tokens | Taildrop | Prob. Taildrop | Index | rFVD↓ | gFVD↓ | gFVD/ rFVD |
|---|---|---|---|---|---|---|---|
| LARP [48] | 1024 | ✗ | ✗ | — | 20.00 | 57.00 | 2.85 |
| VaporTok | 1024 | ✓ | ✗ | sample | 49.45 | 62.34 | 1.26 |
| LARP [48] | 512 | ✗ | ✗ | — | 53.25 | 86.25 | 1.62 |
| VaporTok | 512 | ✓ | ✗ | sample | 81.94 | 93.34 | 1.14 |
| VaporTok | 509 | ✗ | ✓ | argmax | 59.49 | 90.65 | 1.52 |
| VaporTok | 498 | ✗ | ✓ | sample | 53.92 | 80.13 | 1.48 |
| VaporTok | 409 | ✗ | ✓ | pre-sample | 73.01 | 95.41 | 1.30 |

Table 3: Entropy of taildrop probabilities under different GRPO implementation on UCF101 validation set.

| Model | Diversity Reward | Parallel Sampling | TopK Sampling | Entropy |
|---|---|---|---|---|
| VaporTok | — | — | — | 5.11308 |
| VaporTok-GRPO | ✗ | ✗ | ✓ | 0.00642 |
| VaporTok-GRPO | ✗ | ✓ | ✗ | 0.01795 |
| VaporTok-GRPO | ✓ | ✓ | ✗ | 4.11323 |

## 4.1 Video reconstruction & generation comparison

On the UCF-101 class-conditional generation benchmark, we evaluate LARP against video generation approaches—spanning diffusion-based models, masked-language-modeling methods, and autoregressive methods as in [48]. As shown in Table 1, VaporTok achieves competitive performance with other video generators on the UCF-101 dataset even when using significantly fewer tokens. Notably, our VaporTok model shows a much smaller gap between gFVD and rFVD than other AR-based video generators. Besides, after GRPO training, we can further reduce the average latent token count from roughly $50\%$ down to about $30\%$ of the original while still preserving reconstruction and generation quality—thus achieving efficient adaptivity.

## 4.2 Comparison of training techniques

To demonstrate the effectiveness of proposed probabilistic taildrop technique, we make a comparison of different type of training strategies and show the evaluation result in Table 2. For *naive taildrop*, we conducted experiments to test whether taildrop can induce a progressively decreasing importance of tokens along the sequence. From first two rows in Table 2: taildrop causes a substantial performance drop in rFVD, from 20 to 49.45, yet gFVD almost fully recovers the gap introduced by reconstruction and achieve competitive gFVD 62.34 with 57. This shows that taildrop training can partially close the original gap between reconstruction and generation performance by enforcing a semantic-to-detail ordering in the latent space, which greatly reduces error accumulation during autoregressive inference. The same conclusion can also be drawn from the third and fourth rows in Table 2. For *probabilitstic taildrop*, we examine three strategies for obtaining the truncation index from the taildrop probability distribution: taking the argmax, directly sampling, and a pre-sampling variant that only samples from indices before the argmax. Among these strategies, directly sampling from the taildrop probability yields the best performance, achieve the best gFVD 80.13 with a similar token count to baseline methods. The argmax approach lacks variability across different lengths for the same video, thereby missing the core advantage of taildrop—structuring the latent space from semantic to detail. Pre-sampling best reflects this advantage, but it tends to reduce the average token count significantly, which slightly compromises reconstruction and generation quality.

## 4.3 Ablation studies

**Impact of different sampling strategies.** To verify that Parallel Sample GRPO alleviates mode collapse introduced by non-Markovian setting, we compute the average entropy of the taildrop probabilities on UCF101 validation set. The results appear in Table 3: The conclusion that probabilities adjusted by GRPO inevitably become more concentrated can be drawn from the lower entropy compared to that optimized by the prior probability of the first stage as shown at the first row in Table 3. This concentration is an unavoidable consequence of mode collapse introduced by our definition of sequential decision process. However, unlike the complete collapse observed with direct topk sampling, our parallel sampling combined with an exploration reward effectively mitigates the issue.

**Impact of four rewards in Vapor Reward.** During GRPO fine-tuning, we only adjust the taildrop probability without altering the latent-space distribution, our goal is to reduce token usage without degrading generation or reconstruction quality—that is, to achieve an efficiently task-aware adaptive tokenizer.

Firstly, we isolated the individual effects of the generation reward and reconstruction reward. We conducted three ablations: (a) dropping the generation reward, (b) dropping the reconstruction reward, and (c) dropping both. Figure 4 shows that relying solely on the generation reward yields a large boost in generation quality at the expense of reconstruction quality, whereas relying solely on the reconstruction reward greatly improves reconstruction with negligible impact on generation. If both rewards are removed, the model suffers its worst overall performance on both tasks. Notably, using both rewards simultaneously allows steady improvement in both without compromising either. These results confirm that jointly optimizing generation and reconstruction rewards outperforms using either one alone or none at all.

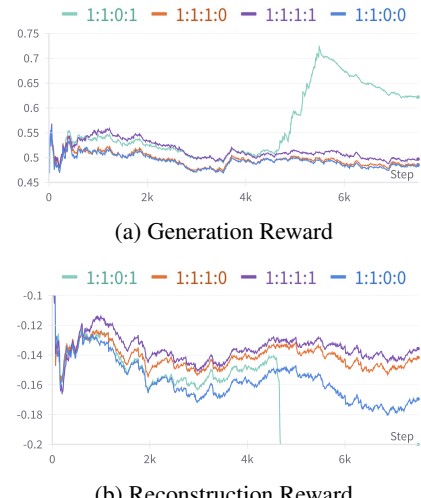

(a) Generation Reward

(b) Reconstruction Reward

Figure 4: The generation (top) and reconstruction (bottom) rewards under different reward weights where the weights order is efficiency, diversity, reconstruction, and generation.

To rigorously assess all four rewards, we carried out a full four-way ablation, removing each reward in turn. The results are summarized in Table 5. Without the efficiency reward, the performance of reconstruction and generation become stronger, but this leads to an increase in the average token cost, violating our efficiency goal. Leaving out the diversity reward produces comparable task performance but causes the taildrop probability to collapse to a few fixed indices, undermining true adaptivity. Omitting

Table 4: Ablation of each rewards where each row "w/o $R$" indicates the model trained without reward $R$. The reported token counts are the average number of tokens used per video on the UCF101 validation set.

| Missing Reward | #Tokens | rFVD↓ | PSNR↑ | gFVD↓ | MSE↓ | ACC↑ |
|---|---|---|---|---|---|---|
| VaporTok-GRPO | 361 | 66.6 | 24.49 | 97.92 | $4.48 \times 10^{-3}$ | 3.70% |
| w/o efficiency reward | 874 | 42.2 | 27.08 | 60.17 | $2.52 \times 10^{-3}$ | 4.44% |
| w/o diversity reward | 325 | 74.9 | 24.23 | 100.94 | $4.77 \times 10^{-3}$ | 3.65% |
| w/o reconstruction reward | 318 | 73.7 | 24.19 | 109.52 | $4.77 \times 10^{-3}$ | 3.60% |
| w/o generation reward | 297 | 77.5 | 24.04 | 113.56 | $4.96 \times 10^{-3}$ | 3.56% |

either the reconstruction or generation reward leads to a performance drop in corresponding tasks. In conclusion, dropping any single reward prevents the model from maintaining reconstruction and generation quality in an efficiently adaptive manner.

## 4.4 The alignment between FVD and reward

To verify the alignment between the final FVD and the reward, the results of reconstruction and generation under both sampling and argmax settings are shown in the Figure 5. It can be observed that rFVD strongly correlates with the reconstruction reward, while gFVD shows a similarly strong correlation with the generation reward. This further supports the validity of our reward design: **the reconstruction MSE serves as a reliable proxy for reconstruction quality, and the top-5 accuracy of the prior model effectively reflects generation quality.**

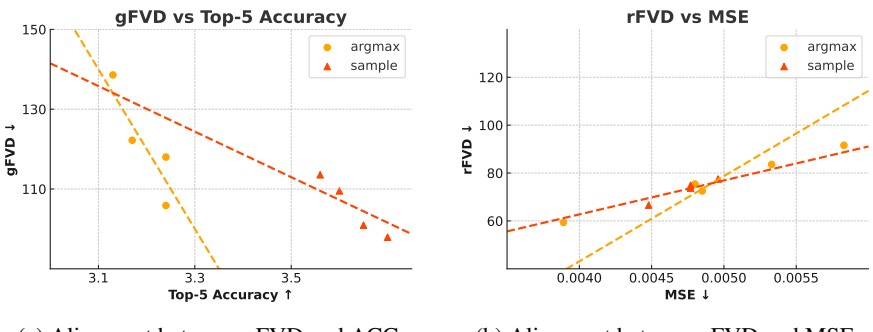

(a) Alignment between gFVD and ACC   (b) Alignment between rFVD and MSE

Figure 5: rFVD and MSE exhibit a strong positive correlation, while gFVD and ACC show a clear negative correlation, indicating the effectiveness and rationality of the proposed reward design.

## 5 Conclusion and future work

We introduce VaporTok, an efficient and adaptive video tokenizer with two key innovations. First, our probabilistic taildrop leverages visual complexity to dynamically determine truncation indexes, preserving semantic-to-detail token structure. Second, we introduce a parallel sample GRPO strategy guided by the Vapor Reward, a unified signal combining token count, reconstruction quality, and generation fidelity, to inject multiple task-related information to VaporTok. Our results show that adaptive tokenization can be effectively learned during training, and demonstrate the effectiveness of GRPO in optimizing tokenizers. However, the present work explores only how to employ GRPO to optimize the truncation probability rather than the entire VAE, and only the generation task is considered as downstream task. Moreover, although various methods have been employed to mitigate mode collapse, the truncation diversity of VaporTok after GRPO training remains notably lower than that after the prior training. Future work will investigate extending GRPO to entire VAE optimization, modeling video tokenization as a complete Markov decision process and applying the framework to understanding [34, 65, 44] and unified generation&understanding scenarios as in [28, 9, 33, 53]

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

# Appendix:

## A  Additional Methodological Details about VaporTok

### A.1  Computation of video spatio&temporal complexity

Given a video tensor $V \in \mathbb{R}^{T \times H \times W \times 3}$, each frame is converted to grayscale via

$$g_t(x, y) = 0.299\, V_{t,x,y,1} + 0.587\, V_{t,x,y,2} + 0.114\, V_{t,x,y,3}, \tag{20}$$

,where $t = 1, \ldots, T$ and $(x, y) \in \{1, \ldots, H\} \times \{1, \ldots, W\}$.

**Spatial Complexity.** Define the empirical pixel-value distribution of frame $t$ as

$$p_t(v) = \frac{1}{HW} \big|\{(x, y) \mid g_t(x, y) = v\}\big|, \quad v = 0, 1, \ldots, 255. \tag{21}$$

The Shannon entropy of frame $t$ is

$$H_t = -\sum_{v=0}^{255} p_t(v) \log_2 p_t(v), \tag{22}$$

and the spatial complexity is the average frame entropy:

$$\mathrm{SC} = \frac{1}{T} \sum_{t=1}^{T} H_t. \tag{23}$$

**Temporal Complexity.** The temporal complexity is defined as the mean absolute difference between consecutive frames:

$$\mathrm{TC} = \frac{1}{(T-1)\, H\, W} \sum_{t=1}^{T-1} \sum_{x=1}^{H} \sum_{y=1}^{W} \big|g_{t+1}(x, y) - g_t(x, y)\big|. \tag{24}$$

### A.2  AR prior in VaporTok

Inspired by the AR prior model introduced in LARP [48], we integrate a similar lightweight autoregressive model into VaporTok as shown in Figure 6. This AR model is designed to make latent tokens more compatible with downstream AR generation tasks, and thus its implementation and associated evaluation metrics can serve as a proxy for downstream generation performance.

Similar to LARP, our lightweight AR model is trained jointly with the VaporTok tokenizer in an end-to-end manner. Specifically, the model takes the quantized latent token embeddings as input, and uses the corresponding codebook IDs as labels. To address the instability caused by the training-inference discrepancy inherent to AR models, Scheduled Sampling Mixing as proposed in [3, 30] is employed.

The key difference is that the prior model used in VaporTok is trained **only on the retained latent tokens after truncation** rather than the whole latent space. Moreover, to enable efficient batchwise training, we adopt the attention masking scheme described in Section A.3 within the transformer blocks of the AR prior model.

### A.3  Attention mask in reconstruction&generation pipeline

**Attention mask for the VaporTok decoder.** During the training of VaporTok, the spatiotemporal complexity of each video sample varies, which results in different taildrop probabilities. Additionally, since sampling is performed over the entire probability distribution during training. These two factors lead to varying truncation positions across samples. Consequently, it becomes infeasible to reconstruct all samples within a batch using a shared decoder input length.To address this issue, we design an adaptive attention mask for the VaporTok decoder to accommodate the variable token lengths caused by the probabilistic taildrop. Specifically, as illustrated in Figure 7(a): For each sample, we construct an individual attention mask: all tokens from decoder queries $M$ are granted

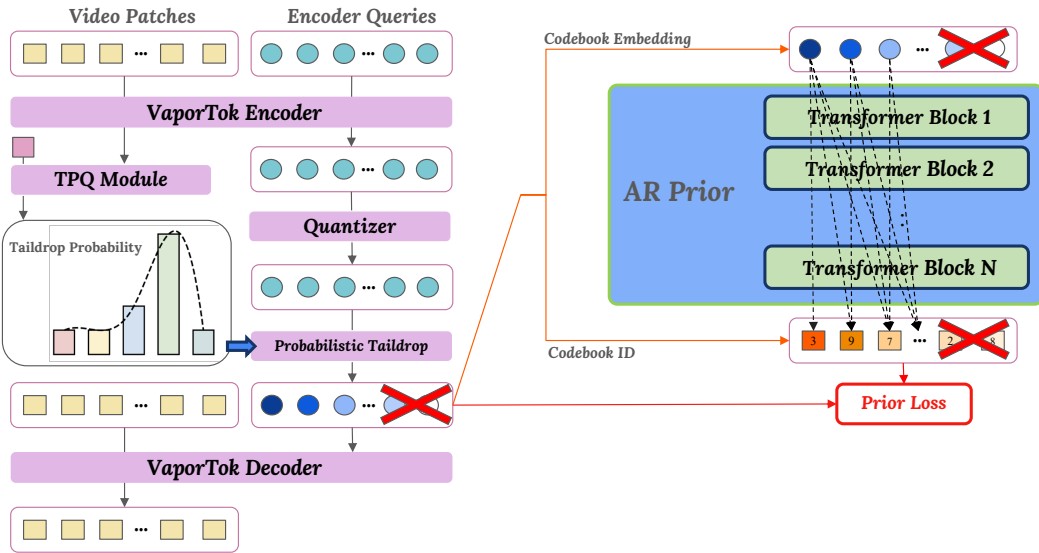

Figure 6: AR prior model in VaporTok.

**full visibility**, while for latent tokens, positions **beyond the truncation point** are masked out to ensure that dropped tokens do not participate in attention computation. This prevents non-informative tokens from interfering with the training process and allows batchwise training of VaporTok.

**Attention mask for the AR prior model.** The AR prior model originally adopts a causal attention mask, which ensures that later tokens do not affect the prediction of earlier tokens. However, if we apply a standard causal mask without modification, tokens before the truncation point can still attend to and influence those after the truncation, which is undesirable. Due to the introduction of taildrop, tokens after the truncation point should not be supervised and influenced by prior tokens during training. To resolve this, we propose a modified attention mask as shown in Figure 7(b).

**Attention mask for the downstream AR generative model.** Since the AR prior model serves as a compact abstraction of the downstream AR generative model, the attention mask used in the downstream AR model is identical to that of the AR prior model, which is also illustrated as Figure 7(b).

### A.4 The complete loss function of VaporTok

$$\mathcal{L}_{\text{rec}} = \lambda_{\text{L1}} \cdot \mathcal{L}_{\text{L1}} + \lambda_{\text{perc}} \cdot \mathcal{L}_{\text{perc}} + \lambda_{\text{GAN}} \cdot \mathcal{L}_{\text{GAN}} + \lambda_{\text{commit}} \cdot \mathcal{L}_{\text{commit}} \tag{25}$$

$$\mathcal{L}_{\text{representation prior}} = -\frac{1}{N} \sum_{i=1}^{N} \log p(y_i \mid x_i) \tag{26}$$

$$\mathcal{L}_{\text{probability prior}} = \text{KL}(P \,\|\, GaussianPrior) \tag{27}$$

$\mathcal{L}_{\text{rec}}$ is comprised of L1 loss, perceive loss, GAN loss, and commitment loss as traditional VQ tokenizer. In Equation 26, $x$ denotes codebook embedding, $y$ denotes codebookid. In Equation 27, $P$ denotes taildrop probability and $GaussianPrior$ denotes prior probability. The complete loss of VaporTok is:

$$\mathcal{L}_{\text{complete}} = \mathcal{L}_{\text{rec}} + \lambda_{\text{rep}} \cdot \mathcal{L}_{\text{rep\_prior}} + \lambda_{\text{prob}} \cdot \mathcal{L}_{\text{prob\_prior}} \tag{28}$$

## B  Supplementary Experiments on Vapor Reward

### B.1  Penalty reward

In our experiment of argmax sampling of probabilistic taildrop, the truncation index always becomes too small to be enough to reconstruct&generate videos, so a reward to punish such truncation is

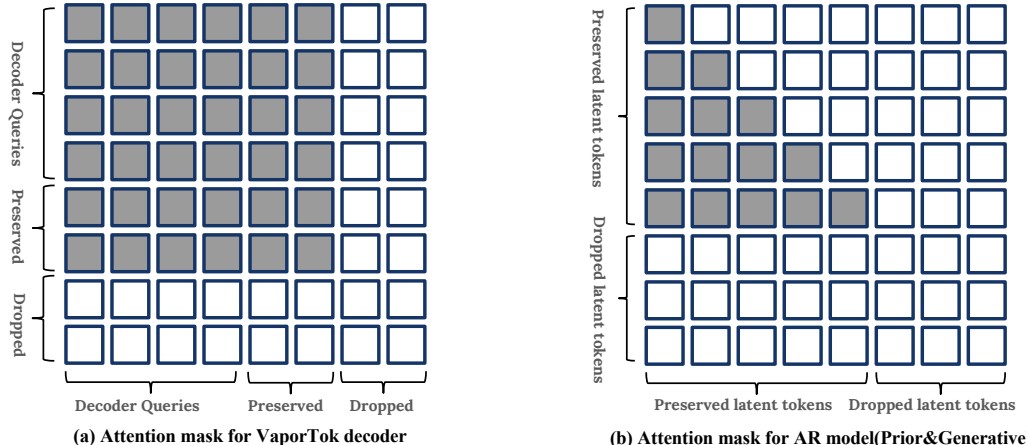

(a) Attention mask for VaporTok decoder      (b) Attention mask for AR model(Prior&Generative)

Figure 7: Attention mask for different parts.

introduced in such senario. Specifically, for the $i$-th path, the penalty reward is defined as:

$$R_{\text{penalty}}^{(i)} = -\sum_{j=1}^{L} \text{Penalty}(I_{i,j}), \quad \text{if } I_{i,j} < N_{\text{threshold}} \tag{29}$$

where $K_{threshold}$ is a manually defined threshold specifying the minimum number of tokens tolerable for reconstructing the video clip and $\text{Penalty}(t_i)$ denotes a penalty function that imposes more punishment when the truncation index becomes smaller. Then we design a five way ablation study in argmax sampling strategy of probabilistic taildrop as shown in Table 5:

Table 5: Ablation of each rewards. Each row "w/o $R$" indicates the model trained without reward $R$. Lower is better for rFVD, gFVD and MSE; higher is better for PSNR and prior top-5 accuracy.

| Missing Reward | #Tokens | rFVD↓ | PSNR↑ | gFVD↓ | MSE↓ | ACC↑ |
|---|---|---|---|---|---|---|
| VaporTok-GRPO | 299 | 72.5 | 24.1 | 118 | $4.85 \times 10^{-3}$ | 3.24% |
| w/o efficiency reward | 577 | 59.4 | 25.6 | 88.46 | $3.89 \times 10^{-3}$ | 3.88% |
| w/o diversity reward | 305 | 75.4 | 24.7 | 105.85 | $4.80 \times 10^{-3}$ | 3.24% |
| w/o reconstruction reward | 272 | 91.6 | 23.3 | 138.58 | $5.83 \times 10^{-3}$ | 3.13% |
| w/o generation reward | 286 | 83.6 | 23.8 | 122.21 | $5.33 \times 10^{-3}$ | 3.17% |
| w/o penalty reward | 58 | 3267 | 10.7 | 3255.59 | $9.45 \times 10^{-2}$ | 2.73% |

## B.2 Effect of reconstruction&generation reward under different weight

We adopt a baseline weighting of 1:1:1:1:1 for the efficiency, penalty, diversity, reconstruction, and generation rewards respectively and then increased the proportion of each of reconstruction&generation rewards. As shown in Figure 8, as the weights for these two rewards grow, their numerical values also increase, which demonstrates that the higher the combined weight on these two performance rewards, the more faithfully the original reconstruction and generation quality is preserved. It is worth noting that if both weights are set too high, performance gains come at the expense of efficiency reward, contradicting our original intent that making tokenizer to be more efficient.

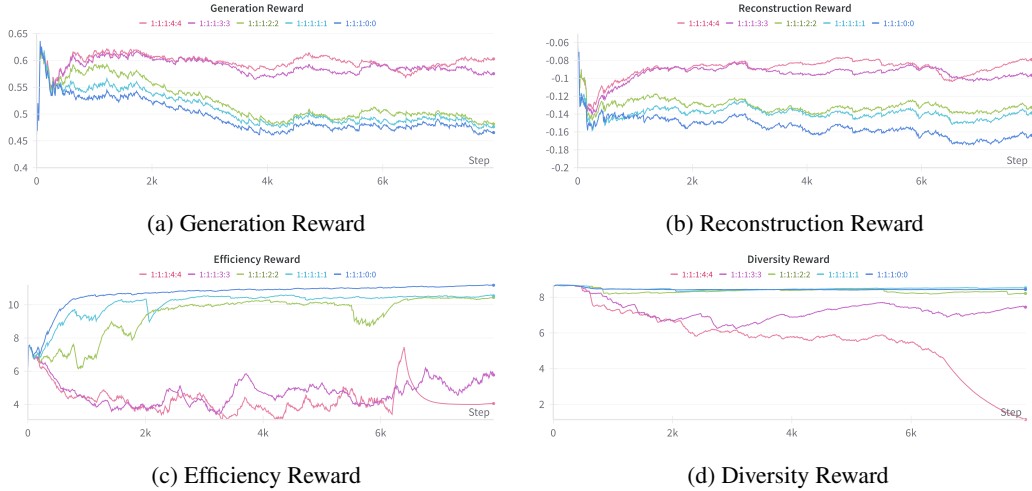

(a) Generation Reward

(b) Reconstruction Reward

(c) Efficiency Reward

(d) Diversity Reward

Figure 8: (a) generation reward (b) reconstruction reward (c) efficiency reward (d) diversity reward of differenct reward weight where the order is efficiency, penalty, diversity, reconstruction, generation.

## C  Analysis about VaporTok

### C.1  Mitigating the three core challenges of autoregressive generation

To mitigate the three key limitations of autoregressive (AR) visual generation, we propose a unified solution within the VaporTok framework:

**Quadratic complexity with long sequences.** We introduce a *sparse but sufficient* token representation by leveraging visual priors to reduce the number of tokens required for generation. Furthermore, we apply GRPO to adaptively compress the token sequence while preserving downstream task performance as much as possible.

**Error accumulation during AR inference.** We propose a *probabilistic taildrop* training strategy that pushes important tokens toward the beginning of the visual representation. As a result, during inference, the model generates the most critical tokens when the accumulation of prediction errors is still minimal, thereby mitigating the impact of error accumulation.

**Training gap between the tokenizer and the AR generator.** To close this gap, we adopt two complementary strategies. First, we refine the latent space following the approach of LARP [48] to make it more suitable for AR generation. Second, we leverage the same AR-aligned information to guide the training of the taildrop query module via GRPO. This enables our adaptive tokenizer to incorporate downstream task constraints into both the adaptivity mechanism and the latent token representation, thus reducing the mismatch between the tokenizer and the generator.

### C.2  Taildrop training strategy for semantic-to-detail representation

The taildrop training strategy is designed to encourage a *semantic-to-detail* organization in the latent token representations. This is primarily achieved through the following mechanism: *for the same video sample, train the model using different numbers of tokens for reconstruction.* As a result, tokens at the beginning of the sequence are exposed to the reconstruction objective more frequently, while later tokens appear less often during training. Furthermore, even when only a small number of tokens are used, the model is still tasked with reconstructing the entire video. This encourages early tokens to encode more global, semantic information that is sufficient for reconstruction.

In addition, it is worth noting the distinction between the two evaluation metrics used in visual reconstruction and generation: rFVD and gFVD. By design, gFVD is consistently worse (higher) than rFVD. This is because rFVD evaluates the reconstruction quality using ground-truth codebook ID directly obtained from the encoder, whereas gFVD evaluates the generation quality using codebook ID predicted by the autoregressive model. *Briefly, the only difference between rFVD and gFVD is that*

*reconstruction relies on ground-truth ID, while generation depends on autoregressively predicted ID, making the gap between gFVD and rFVD a direct indicator of the degree of AR error accumulation.*

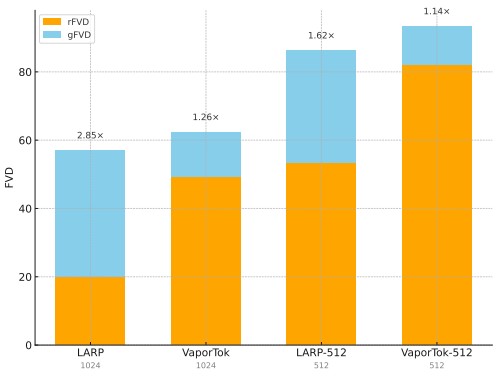

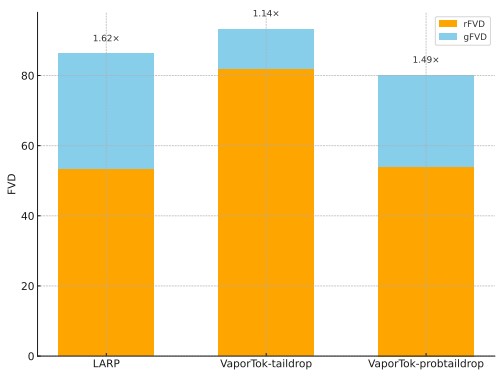

(a) Comparison between LARP and VaporTok with naive taildrop under 1024 and 512 token budgets

(b) Comparison between no technique, naive taildrop, and probabilistic taildrop under 500 token budgets

Figure 9: (a) Naive taildrop reduces the gap between rFVD and gFVD, but it leads to a drop in reconstruction performance, which in turn results in degraded generation performance. (b) Probabilistic taildrop, by incorporating a visual prior, avoids the reconstruction performance degradation caused by taildrop, while preserving its original ability to reduce the gap between rFVD and gFVD.

To quantify this effect, we report the **gFVD/rFVD ratio** as shown in Table 2 of the main paper and Figure 9a to measure the discrepancy between reconstruction and generation. We conduct experiments under both 1024-token and 512-token settings. The results show that with taildrop enabled, the gap between gFVD and rFVD becomes smaller, demonstrating the effectiveness of taildrop in mitigating AR error accumulation; specifically, the gFVD/rFVD ratio decreases from 2.85 to 1.26 under the 1024-token setting, and from 1.62 to 1.14 under the 512-token setting.

## C.3 Difference between naive taildrop and probabilistic taildrop

The distinction between naive and probabilistic taildrop primarily manifests in training and inference:

**Training:** While naive taildrop uses uniform sampling for truncation during training, probabilistic taildrop samples from a learned, prior-informed distribution. (We also investigate three specific sampling strategies under the probabilistic framework.)

- *Truncation by argmax of taildrop probability.* The sequence is truncated at the index corresponding to the maximum value in the taildrop probability distribution. While simple, this approach always uses the same number of tokens for a given input, limiting the semantic-to-detail effect.

- *Sampling from the full taildrop probability.* The truncation index is sampled from the entire taildrop probability distribution. This allows different truncation lengths for the same input and leads to strong reconstruction performance.

- *Sampling indices before the argmax index.* We sample a truncation index from the region before the argmax position, based on the taildrop probability. This also enables varying token counts across samples, though typically results in fewer tokens and slightly degraded reconstruction quality.

**Inference:** During inference, naive taildrop uses the full set of tokens for decoding (which is the default inference mode used for all reported results in this paper. Alternatively, one may adopt a threshold-based token selection strategy, as in [57]). In contrast, probabilistic taildrop performs decoding using all tokens preceding the argmax index of the taildrop probability distribution, achieving an adaptive number of tokens based on input complexity.

It is worth noting that, among the three sampling strategies for probabilistic taildrop during training (from argmax to sample and then to pre-sample), the degree of *semantic-to-detail* structure in the

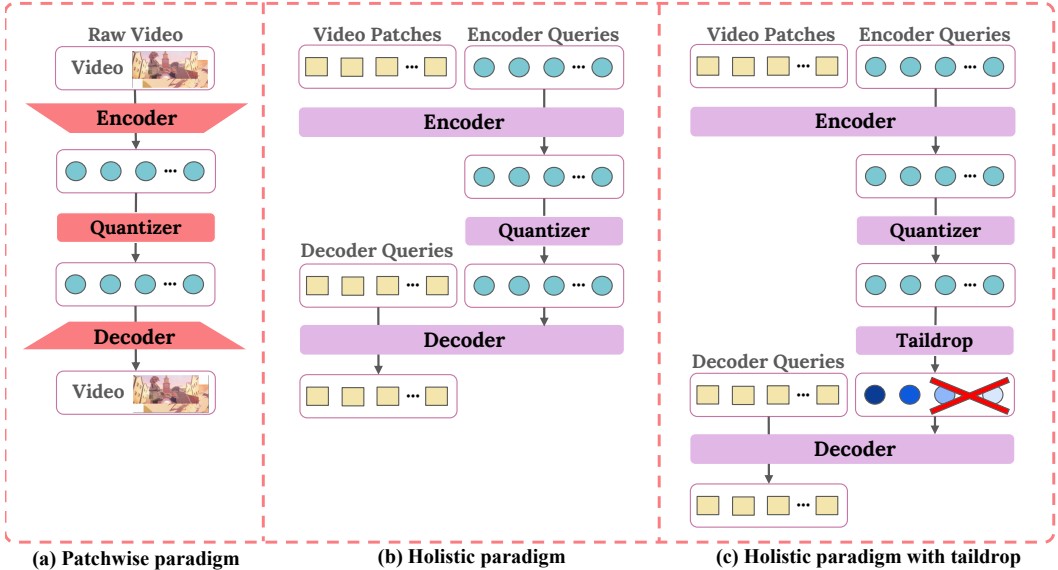

| (a) Patchwise paradigm | (b) Holistic paradigm | (c) Holistic paradigm with taildrop |

Figure 10: Different paradigm of visual tokenizer: (a) The patchwise-token paradigm typically represents each token as encoding information from a specific spatial region and usually adopts a CNN as the backbone.(b) The holistic-token paradigm is not constrained by fixed spatial positions and can flexibly adjust the information each token represents based on the training strategy. (c) Taildrop is a training technique commonly used in the holistic-token paradigm, enabling the token sequence to exhibit a semantic-to-detail property.

tokens used during inference gradually increases because of the frequency of sampling index before argmax index is gradually frequent. In other words, these strategies increasingly mitigate AR error accumulation. This trend is also validated by the experimental results reported in Table 2 of the main text as the gFVD/rFVD ratio becomes smaller.

Furthermore, while naive taildrop helps narrow the gap between reconstruction and generation (rFVD vs. gFVD), its uninformed dropping during training—without accounting for visual priors—results in compromised reconstruction quality. In contrast, our probabilistic taildrop incorporates a learned prior distribution, maintaining competitive reconstruction performance (rFVD) and achieving superior generation quality by alleviating AR error accumulation.

### C.4 Different visual tokenizer paradigm

Visual tokenizers can be classified according to various criteria, and one particularly informative distinction is how they map visual patches to latent tokens, yielding two families: Patchwise-Token and Holistic-Token.

In the Patchwise-Token paradigm as depicted in Figure 10(a) , each learned token corresponds one-to-one with a visual patch, thereby preserving the spatial structure imposed. Briefly, given a video input $V \in \mathbb{R}^{T \times H \times W \times 3}$, the encoder outputs a downsampled feature map

$$Z = \text{Enc}(V) \ \in \ \mathbb{R}^{\frac{T}{f_T} \times \frac{H}{f_H} \times \frac{W}{f_W} \times D}, \tag{30}$$

where $f_T, f_H, f_W$ are the temporal and spatial downsampling factors. The reconstructed video is then obtained as

$$\widehat{V} = \text{Dec}(Z) \in \mathbb{R}^{T \times H \times W \times 3}. \tag{31}$$

In the Holistic-Token paradigm as depicted in Figure 10(b), each latent token may assume different semantic roles depending on the training strategy, resulting in a more flexible representational scope. Different from Patchwise-Token paradigm mainly depends on 3D CNN as the backbone, Holistic-Token paradigm usually employs Transformer blocks as the backbone, so a simple patch embedding layer is needed to be conducted to $V \in \mathbb{R}^{T \times H \times W \times 3}$

$$P = \text{Patchify}(V) \in \mathbb{R}^{(\frac{T}{f_T} \times \frac{H}{f_H} \times \frac{W}{f_W}) \times D}, \tag{32}$$

and then $P$ will be concated with $K$ learnable query tokens $Q \in \mathbb{R}^{K \times D}$ and the combined sequence will be passed into the encoder:

$$Z_P \oplus Z_Q = \text{Enc}(P \oplus Q) \in \mathbb{R}^{(\frac{T}{f_T} \times \frac{H}{f_H} \times \frac{W}{f_W} + K) \times D}, \tag{33}$$

where $\oplus$ denotes concatenation and $Z_P, Z_Q$ denotes the represatation of $P$, $Q$ after encoder respectively. During detokenization, $M \in \mathbb{R}^{(\frac{T}{f_T} \times \frac{H}{f_H} \times \frac{W}{f_W}) \times D}$, the decoder query of the same shape as the video patches $P$, will be concatenated with $Z_Q$ and passed to the decoder to rencostruct the input video.

$$\widehat{V} = \text{Dec}(M \oplus Z_Q) \in \mathbb{R}^{T \times H \times W \times 3}. \tag{34}$$

It is worth noting that *the taildrop training strategy is typically employed in holistic-token tokenizers*. This is primarily because the query representation is not constrained by fixed spatial regions, allowing it to flexibly adapt to different training objectives and strategies. Moreover, such flexibility enables the model to achieve effective representation learning with less training data.

## D   Supplementary Related Work about Fixed-length Visual Tokenizer

Visual tokenizers for understanding tasks typically rely on contrastive learning, for example: CLIP[34] trains paired image and text encoders with an InfoNCE contrastive loss over matched image–caption pairs, enabling strong zero-shot transfer across diverse vision task; SigLIP[65] replaces CLIP's softmax-based InfoNCE loss with an independent pairwise sigmoid loss, removing the need for global normalization and scaling more efficiently to very large or small batch sizes. TULIP[44] augments CLIP-style pretraining with generative data augmentation and unified image–image, text–text, and image–text contrastive objectives plus reconstruction regularization to learn fine-grained visual features without sacrificing semantic alignment.

Whereas visual tokenizers designed for generation usually employ VAE-based architectures: VQ-VAE[47] first introduce vector quantization into VAE, transforming data from continuous spaces into discrete tokens to simplify modeling and circumvent issues of "posterior collapse" in VAE framework; VQ-GAN[12] improves image reconstruction quality by introducing adversarial loss and using a Transformer for autoregressive visual generation; FSQ[29] projects representations into a lower dimensional space for quantization into fixed values, while its variant LFQ[62, 27] further simplifies the process by using binary quantized representations. This new kind of quantization method effectively enhances the AR generation paradigm by dramatically enlarging the vocab size and improving the encoding efficiency. Beyond these patch-to-token VAEs, there are also VAEs that learn holistic tokens, such as TiTok[63] and LARP[48], by compressing visual information into a holistic query, they eliminate the patch-to-token correspondence constraint, yielding a more flexible architecture with inherent compression potential.

Recently, along with the rapid development of the unified model[52, 54, 53, 69], there are also several visual tokenizers designed for both generation and understanding, such as TokenFlow[33], SemHiTok[9] and UniTok[28]. These works focus on unifying visual generation and understanding within a single visual tokenizer by employing discrete representations and hierarchical or multi-codebook strategies, addressing the differing requirements in token granularity and semantic level between generation and understanding tasks.

While the above methods have shown impressive results for static images, extending visual tokenizers to video requires capturing both temporal continuity and spatial detail, presenting new challenges for tokenizer architectures and training paradigms. Recent work addresses this by integrating temporal modeling[49], diffusion-guided reconstruction[16], hierarchical codebooks[68], and coordinate-based patch schemes[20] to efficiently compress and faithfully reconstruct long video sequences.

## E   Visualization

We present visualizations of VaporTok's reconstruction and final class-based generation results, as shown in the Figure 11 and Figure 12.

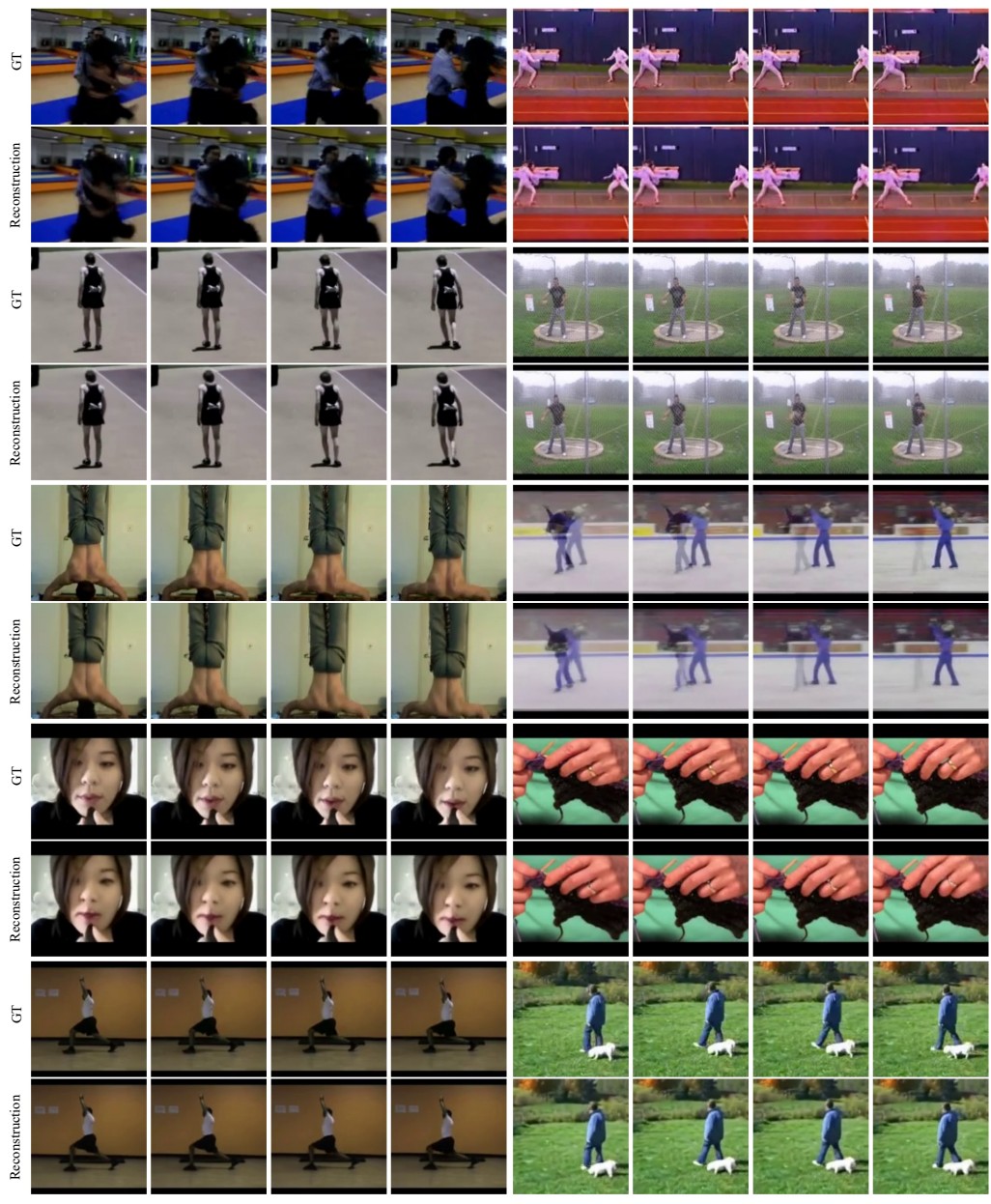

Figure 11: Video reconstruction on UCF101.

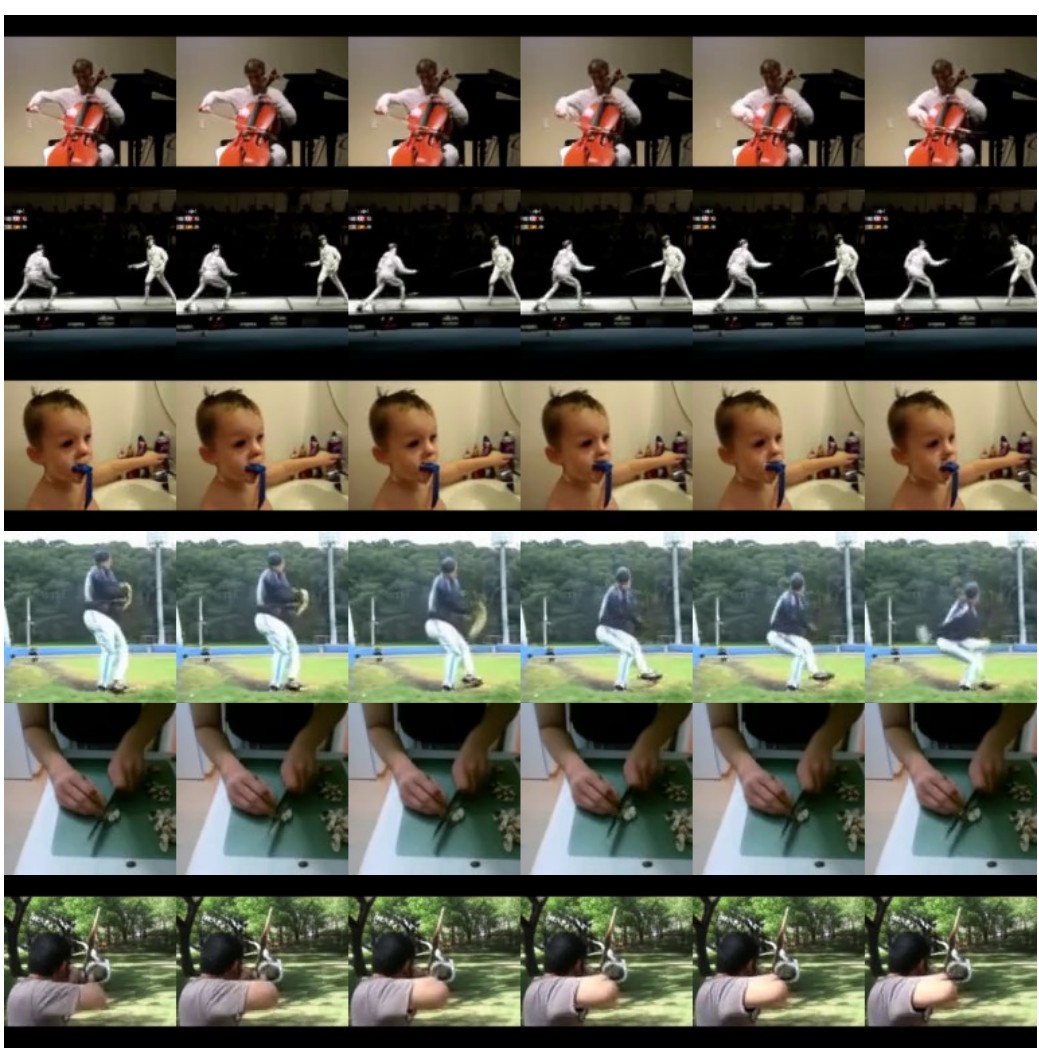

Figure 12: Class-based video generation on UCF101.

# F    Broader Impacts

Our Adaptive Video Tokenizer is designed exclusively for autoregressive video generation tasks, and is not intended for video understanding or classification; by allocating fewer tokens to low-complexity videos and more to high-detail videos, our method reduces computational and bandwidth costs for real-time generative applications (e.g., interactive video editing, virtual content creation); it enables fast, adaptive video synthesis for artistic tools and educational simulators, lowering barriers for non-expert users to generate high-fidelity video content; it facilitates deployment of generative video models on edge devices (e.g., AR/VR headsets, mobile phones) by reducing token sequence length and inference latency; however, improved efficiency in video generation could be misused to produce highly realistic deepfake videos, exacerbating misinformation campaigns; although not designed for recognition, the underlying tokenizer could be adapted to generate misleading synthetic footage for surveillance evasion or identity spoofing; training on unbalanced datasets may lead the tokenizer to allocate token budgets unevenly, causing generative artifacts that disproportionately affect certain demographics; while inference is more efficient, training the dual-branch model remains GPU-intensive, contributing to carbon emissions.

# G   Safeguards

To mitigate potential misuse and harms, we will release a detailed model card specifying that use for deceptive or harmful video synthesis (e.g., deepfakes) is prohibited; distribute weights under a non-commercial, no-derivatives license (e.g., CC BY-NC-ND) and/or via an API with rate limits rather than open weight download; embed imperceptible watermarks in generated videos for provenance tracking and provide a companion detection model to flag synthetic content; maintain a public issue tracker for misuse reports and regularly update the model to address discovered biases or vulnerabilities; publish training logs, compute cost estimates, and carbon-emission metrics to inform users of environmental impact.

