# OpenReview forum: "VaporTok: RL-Driven Adaptive Video Tokenizer with Prior & Task Awareness"
_NeurIPS.cc/2025/Conference — NeurIPS 2025 poster_

### Official Review · Reviewer_T8WN · 2025-07-01

**Clarity:** 2
**Significance:** 3
**Originality:** 3
**Rating:** 4
**Confidence:** 2

**Summary:**

This paper proposes an adaptive video tokenizer that produces varying visual tokens according to input complexity. To achieve this adaptivity, two core designs are introduced: (1) a taildrop mechanismm that truncates video token according to its complexity, (2) a group relative policy optimization (GRPO) module that learns visual tokenization as a sequential decision process. The proposed tokenizer is evaluated in both video reconstruction and generation experiments, demonstrating its effectiveness.

**Questions:**

Please ref to the weaknesses section.

**Ethical Concerns:**

["NO or VERY MINOR ethics concerns only"]

**Final Justification:**

The authors have addressed my concerns. Therefore, I maintain my original rating of acceptance.

**Limitations:**

yes

**Paper Formatting Concerns:**

The paper does not have a major formatting issue.

**Quality:**

2

**Strengths And Weaknesses:**

Strength:

* This paper studies an imortant problem of adaptive video tokenization. This problem is critical because of the high requirement of video tokenization despite varying complexity of videos.
* Experimental results show the effectiveness of their proposed method.

Weaknesses:

* Comparison between reinforcement learning and supervised finetuning is insufficient. For example, reinforcement learning has this reward hacking issue and has limited generalizability. Meanwhile, supervised finetuning cannot resolve back-side sears. The authors could conduct additional ablation study to demonstrate the advantages of their method.
* Hyperparameter sensitivity is not well discussed. While the paper introduces multiple weights within the reward model (in line 244 of page 7), it would complicate the framework and the same set of hyperparameters might not generalize to different scenarios .
* Lack of human study. Despite that automatic evaluation metrics like PSNR shows some advantages of the proposed method, many video issues such as blurriness and artifacts cannot be fully reflected through automated metrics. Therefore, the authors could conduct additional user study to validate the effectiveness of their model.

---

> ### Author Rebuttal · Authors · 2025-07-31
>
> We appreciate the reviewer’s constructive feedback and have provided our responses below.
>
> ---
>
> ### **Q1: Comparison between reinforcement learning and supervised finetuning.**
>
> To address the reviewers’ concerns, we constructed an SFT dataset using the truncation positions derived from first-stage video complexity prior and directly optimized VaporTok’s taildrop probability. In order to ensure a fair comparison, we adjusted the token count for VaporTok-SFT to be comparable to that of VaporTok-GRPO. The comparative results are as follows:
>
> |  | token num(Avg.) | rFVD | gFVD | entropy |
> | --- | :---: | --- | --- | --- |
> | VaporTok | 498 | 53.9 | 80 | 5.11308 |
> | VaporTok-SFT | 361 | 69.4 | 107 | 5.09822 |
> | VaporTok-GRPO | 361  | 66.6 | 98 | 4.11323 |
>
> It can be seen that, under the same token budget, VaporTok-SFT underperforms VaporTok-GRPO by a considerable margin. The main reason is that during SFT, the gradients from reconstruction and generation cannot be back-propagated to the taildrop probability module, so the taildrop probability is never optimized for either task and just aligns with data prior of the first stage.
>
> By contrast, VaporTok-GRPO explicitly trades off efficiency against reconstruction & generation, allowing further token-count compression at the cost of only a slight drop in reconstruction/generation performance compared to baseline. In summary, GRPO offers two main advantages over SFT:
>
> 1. **Gradient propagation for task objectives.** GRPO circumvents the non-differentiability issue by incorporating reconstruction and generation signals into reward function of GRPO. SFT cannot do this, since one cannot label whether a sampled token sequence benefits generation, and the sampling operation itself blocks precise backpropagation of reconstruction gradients.
> 2. **Parallel, stable training and rapid convergence.** GRPO trains on many video clips simultaneously, inherently improving stability and speed of convergence (see Appendix B.2, Figure 3: the reward stabilizes after ∼8 K steps).
>
> Finally, to address the reviewers’ concern about reward hacking observed in reinforcement learning (i.e., mode collapse, where different videos’ taildrop probabilities peak at the same index), we employ two countermeasures—stochastic parallel sampling and a diversity reward—and Table 3 in the main paper confirms their effectiveness via ablation.
>
> ---
>
> ### **Q2: The discussion about sensitivity and generalization about hyperparameters of the weight of different components in Vapor Reward**
>
> The purpose of this hyperparameter is simply to rescale the four rewards into comparable ranges so that they can be effectively traded off during the GRPO training.
>
> For example, when applying GRPO in the LLM or VLM domains, all rewards are typically constrained to [0, 1].  Likewise, we map our model-based rewards (reconstruction and generation) to [−1, 1] and our rule-based rewards (efficiency and diversity) to [0, 10].  We chose these ranges because, without adjusting the video latent-space distribution during GRPO, the model-based rewards fluctuate much more strongly than the rule-based ones—assigning them a smaller numerical span thus stabilizes training.
>
> After normalizing the scales, in Figure 4 of Section 4.3 and Figure 3 of Appendix B.2 we performed a simple sensitivity analysis on the two most volatile model-based rewards.  We varied their weights to study the impact on final performance, testing settings from 1:1:1:1 → 1:1:1:0 → 1:1:0:1 → 1:1:0:0, and observed that both rewards are indispensable.  Moreover, increasing their weights (from 1:1:1:1 → 1:1:2:2 → 1:1:3:3 → 1:1:4:4) consistently improves reconstruction and generation quality.  These analyses confirm not only the effectiveness of our reward design but also that, even when an order of magnitude smaller than the rule-based rewards, the model-based terms still exert strong control over GRPO training.
>
> As for the generalizability of this hyperparameter: different videos do not produce order-of-magnitude differences in their computed rewards.  We sampled 10 % of the Kinetics-600 validation set and evaluated the rewards with our final VaporTok-GRPO; the mean values (below) show no such magnitude disparities. Thus, the same hyperparameters can be applied to different datasets without significant modification.
>
> | Data | UCF101 | K600 sub set  |
> | --- | :---: | :---: |
> | Efficiency reward | 10.89 | 10.20 |
> | Diversity reward | 8.48 | 8.56 |
> | Reconstruction reward | -0.162 | -0.042 |
> | Generation reward | 0.460 | 0.777 |
>
> ---
>
> ### **Q3: Human study to validate the effectiveness of VaporTok.**
>
> For reconstruction, we used the same 20 videos and reconstructed them with LARP (512 tokens), VaporTok (avg. 534 tokens), ElasticTok (1024 tokens), and OmniTokenizer (1280 tokens), comparing each against the ground truth.  Participants saw four options and could choose two reconstrction results per video (the first choice as most preferred, the second as next preferred).  Eighteen people participated in the test.
>
> Below are the voting results, which show that VaporTok, even with a token count comparable to or significantly lower than other methods, still outperforms them.
>
> | Method | First choice | First choice ratio | Second choice | Second choice ratio | Total choice | Overall ratio |
> | --- | --- | --- | --- | --- | --- | --- |
> | VaporTok(Avg. 534) | 125 | 34.7% | 114 | 31.6% | 239 | 33.19% |
> | LARP(512) | 113 | 31.3% | 105 | 29.1% | 218 | 30.27% |
> | OmniTokenizer(1280) | 115 | 31.9% | 130 | 36.1% | 245 | 34.02% |
> | ElasticTok(1024) | 7 | 1.9% | 11 | 3.0% | 18 | 2.5% |
> | Total | 360 | \ | 360 | \ | 720 | \ |
>
> For generation evaluation, we randomly selected 10 classes from UCF101. For each class, we generated one video with OmniTokenizer, LARP, and VaporTok. Participants then chose which model produced the best result. Eighteen people took part in the test. It can be seen that, with a comparable token count, our method achieves superior user preference.
>
> | Method | Choice | Choice ratio |
> | --- | --- | :---: |
> | OmniTokenizer(1280) | 5 | 2.7% |
> | LARP(512) | 69 | 38.3% |
> | VaporTok(Avg.525) | 106 | 58.8% |
> | Total | 180 | \ |

---

> > ### Comment · Reviewer_T8WN · 2025-08-05
> >
> > The authors have addressed my concerns. I maintain my recommendation for acceptance.

---

> > > ### Author Response · Authors · 2025-08-05
> > >
> > > We sincerely appreciate the reviewer’s positive feedback and insightful suggestions, which have greatly strengthened our work.

---

### Official Review · Reviewer_8AKU · 2025-07-03

**Clarity:** 3
**Significance:** 2
**Originality:** 2
**Rating:** 4
**Confidence:** 5

**Summary:**

This paper proposes an adaptive video tokenizer named VaporTok. It proposes a taildrop mechanism that learns a truncation index sampling distribution conditioned on visual complexity of the video. By doing this,  the decoder reconstructs videos at adaptive token lengths, allocating more tokens to complex videos and fewer to simpler ones. Further, the authors reformulate the visual tokenization pipeline as a sequential decision process and enables metrics-aware adaptive tokenization across diverse objectives. Extensive experiments show the effectiveness of the method.

**Questions:**

see weakness.

**Ethical Concerns:**

["NO or VERY MINOR ethics concerns only"]

**Final Justification:**

The author response has addressed most of my concerns (baseline comparison, model complexity). So i raise my score to 4.

**Quality:**

2

**Strengths And Weaknesses:**

Strengths:

+ The paper proposes an adaptive video tokenization technique adjusting token count based on visual complexity for more efficient sequences.

+ The paper formulates video tokenization as a sequential decision process and uses RL to optimize the tokenizer. The idea is novel.

+ The paper is well written and easy to read.

Weakness:

- My main concerns are about the performance comparison. For example, the Authors can consider compare with more baseline models which also use adaptive tokenizers [R1-R4].

- More analysis on model complexity needs to be given. For example, will the introduction of adaptive encoding based on reinforcement learning significantly increase the computational complexity and time delay of the model? And will it make the model significantly more difficult to optimize and train to converge?

- More training details shoud to be provided. For example, how much time/GPU hours does it take to train the model? And the use of GPU cards or clusters. Besides, a comparison of the training resources with the baseline models should also be provided.

[R1] Visual lexicon: Rich image features in language space. 2024

[R2]  Elastictok: Adaptive tokenization for image and video. 2025.

[R3] Flextok: Resampling images into 1d token sequences of flexible length. 2024.

[R4] One-d-piece: Image tokenizer meets quality-controllable compression. 2025.

---

> ### Author Rebuttal · Authors · 2025-07-30
>
> We thank reviewer for the constructive comments. We provide our feedbacks as follows.
>
> ---
>
> ### **Q1:  Compare with more baseline models which also use adaptive tokenizers.**
>
> We have already discussed adaptive visual tokenizer approaches such as Vilex[1], FlexTok[2], One-D-Piece[3], and ElasticTok[4] in the related work section of paper; any additional methods we have not mentioned will be included in future versions.
>
> Vilex, Flextok, and One-D-Piece are adaptive **image** tokenizers and therefore cannot be directly compared to **video** tokenizers under a single unified metric. Moreover, ElasticTok only addresses the reconstruction task and does not include a generative component, so we compare it solely on reconstruction performance. To broaden our evaluation, we additionally include Cosmos-Tokenizer-DV[5] and a contemporaneous method, AdapTok[6]. The comparison results are shown below:
>
> | Method | Token(Avg.) | rFVD |
> | --- | --- | --- |
> | ElasticTok | 1024 | 390 |
> | ElasticTok | 2048 | 93 |
> | Cosmos-Tokenizer-DV | 1280 | 140 |
> | AdapTok | 512 | 60 |
> | LARP | 512 | 53 |
> | VaporTok | 498 | 54 |
>
> For AR-style downstream generation tasks, we additionally incorporate Video-LaVIT[7] to complement the gFVD metric. The comparison results are as follows:
>
> | Method | Token(Avg.) | gFVD |
> | --- | --- | --- |
> | MAGVIT-AR | 1024 | 265 |
> | MAGVIT-v2-AR | 1280 | 109 |
> | TATS | 1024 | 332 |
> | OmniTokenizer | 1280 | 191 |
> | CogVideo | 2065 | 626 |
> | Video-LaVIT | 1024 | 281 |
> | LARP | 1024 | 57 |
> | LARP | 512 | 86 |
> | VaporTok | 498 | 80 |
>
> [1] Visual lexicon: Rich image features in language space
>
> [2] Flextok: Resampling images into 1d token sequences of flexible length
>
> [3] One-d-piece: Image tokenizer meets quality-controllable compression
>
> [4] Elastictok: Adaptive tokenization for image and video
>
> [5] Cosmos-Tokenizer: A suite of image and video neural tokenizers
>
> [6] Learning Adaptive and Temporally Causal Video Tokenization in a 1D Latent Space
>
> [7] Video-LaVIT: Unified Video-Language Pre-training with Decoupled Visual-Motional Tokenization
>
> ---
>
> ### **Q2-1: Analysis of the computational complexity of probabilistic taildrop during both training and inference.**
>
> During training, VaporTok learns a Taildrop Probability Query Module (22 M parameters) in addition to the original VAE (173 M parameters). This does introduce extra overhead, but only increases the total training parameters by 12.7%. The module is trained in both stage1 and stage2.
>
> During inference:
>
> - The Taildrop Probability Query Module runs in parallel with token discretization, adding computation but without noticeably increasing inference time.
> - In the decoder, because VaporTok decodes a shorter sequence than the full length, for single-sample reconstruction one can simply truncate (rather than mask) the input before feeding it to the decoder, achieving baseline-equivalent speed.
>
> We evaluated inference time on the UCF101 validation set with a batch size of 1, averaging over five runs. The results are as follows:
>
> |  | VaporTok | VaporTok(directly truncate) | Baseline（LARP） |
> | :---: | :---: | :---: | :---: |
> | Encoder time(bs=1) | 6.7ms | 6.3ms | 6.3ms |
> | Decoder time(bs=1) | 7.3ms | 4.0ms | 3.8ms |
>
> ---
>
> ### **Q2-2: Will the introduction of adaptive encoding based on reinforcement learning make the model significantly more difficult to optimize and train to converge?**
>
> Adaptive encoding based on reinforcement learning does not make the model more difficult to optimize and train to converge. As shown in Appendix C.1, our results demonstrate that after roughly 8,000 steps, the reward values settle into a stable range, so convergence issues do not arise. We analyzed the cause and found that having a good initialization distribution before GRPO is critically important.
>
> If one trains from scratch and simultaneously optimizes both the truncation distribution with GRPO and the latent‐space distribution, convergence can indeed be problematic.
>
> However, VaporTok initializes with a pre-trained weights, at which point the VAE has already learned a strong latent‐space representation. On this foundation, VaporTok learns the taildrop probability from the data prior while concurrently adjusting the latent‐space representation to follow a semantic-to-detail ordering. The data prior is more stable and provides an excellent reference for GRPO; moreover, the KL-divergence term in GRPO’s objective ensures that optimization remains centered around the taildrop probability of first stage, thereby enhancing training stability.
>
> ---
>
> ### **Q3: More training details should to be provided.**
>
> We have already detailed the training hardware and durations in Appendix E; we now summarize them as follows:
>
> - **VaporTok-Stage1:** VaporTok initializes with pre-trained weights of LARP and traines for 30 epochs on the UCF101 and K600 datasets, which requires 90 hours on 8 A100 GPUs.
> - **VaporTok-Stage2:** The GRPO training process uses the UCF101 dataset for a single epoch, which takes 3 hours on a single A100 GPU.
> - **Generation:** The generation task is trained on the UCF101 dataset for 3000 epochs, which takes 40 hours on 8 A100 GPUs.
>
> The baseline model(LARP) is trained for 150 epochs on the UCF101 and K600 datasets and his generation task is also trained on the UCF101 dataset for 3000 epochs, which is as same as ours. However, LARP does not specify the GPU hardware used or the precise duration of training, so we cannot compare training durations with it.

---

> > ### Comment · Reviewer_8AKU · 2025-08-03
> >
> > Thanks for the author's response. I suggest that the authors include these analyses into their revision. I will raise my voting.

---

> > > ### Author Response · Authors · 2025-08-03
> > >
> > > We are very honored by the reviewer’s recognition of our work and deeply grateful for the final decision to raise the score. We will incorporate the points discussed in the rebuttal into the final version.

---

### Official Review · Reviewer_FNXp · 2025-07-03

**Clarity:** 3
**Significance:** 3
**Originality:** 4
**Rating:** 4
**Confidence:** 4

**Summary:**

This paper innovatively proposes an adaptive video tokenizer that adjusts the token count:
1) A probabilistic taildrop module is learned by aligning with a predefined explicit video complexity prior.
2) In the second stage, the probabilistic taildrop layer is further optimized using reinforcement learning with GRPO, where the probabilistic taildrop module serves as the policy model and is optimized according to multiple rewards.

**Questions:**

See weakness above

**Ethical Concerns:**

["NO or VERY MINOR ethics concerns only"]

**Final Justification:**

I'm satisfied with the response and keep my original rating.

**Limitations:**

Yes

**Quality:**

3

**Strengths And Weaknesses:**

Strengths
1. The paper is well-written and easy to follow. To facilitate the shift from fixed-rate to adaptive downsampling, the authors novelty parameterized and optimized the truncation probabilistic taildrop supervised by a visual prior, then further used GRPO to optimize the truncation of the entire video clips as  a sequential decision process.

2. The experimental results on UCF101 and K600 show that VaporTok+GRPO reduces token length by 75%, demonstrating the effectiveness of the adaptivity token adjusting method.


Weaknesses
1. The expression in abstract line 19, "matches and outperforms fixed-rate baselines and naive taildrop while using fewer tokens" is somehow unclear regarding which model is referred to as the "fixed-rate baseline" and "naive taildrop" in Table 1.  From my review of the codebase, it appears that the implementation closely follows the open-sourced LARP model, so I assume the "fixed-rate baseline" refers to LARP. However, VaporTok uses lossy token truncation compression, which introduces a gap compared to LARP in rFVD, from 20 to 53.9(VaporTok) or 66.6(VaporTok-GRPO). The authors do not discuss in the experiment whether this gap is significant or minor.  Simply saying "competitive" is not sufficient.

2. In line 276, it says, “From the first two rows in Table 2: taildrop causes a substantial performance drop in rFVD, from 20 to 49.45.”   I’m confused about why VaporTok with Taildrop still maintains a token length of 1024 same as the LARP baseline.  Could you clarify what exactly is being done here and where the performance loss comes from? Specifically, what is the operation involved, and how does it lead to this drop in performance with the same token length?

3. It would be better to include a discussion on the visualization of the Taildrop Probability Query Module after GRPO at different truncation points across various video clips.  Specifically, it is unclear which videos adopted longer or shorter truncation points, and whether the visualizations align with prior assumptions.   This analysis may be beneficial to investigate whether cold-starting the module using the current visual prior is both expected and necessary

---

> ### Author Rebuttal · Authors · 2025-07-30
>
> We thank the reviewer for acknowledging the novelty of our method and the effectiveness of its efficiency. Below, we provide our responses to the reviewer’s questions.
>
> ---
>
> ### **Q1: What do “fixed-rate baseline” and “naive taildrop” refer to in the abstract of the paper?**
>
> The terms 'fix-rate baseline' and 'naive taildrop' mentioned in the abstract refer to the third and fourth rows in Table 2 of the main paper, respectively. Our probability taildrop experiment was configured with a maximum token length of 1024. However, since we utilize adaptive token count during both training and inference, the actual average number of tokens used is less than 1024, averaging around 500 tokens. The comparison of LARP in table1 of main paper is not a fair comparison, as LARP is using a larger number of tokens, so we have constructed a fairer comparison below (with token counts of around 500). We mitigated the shortcoming of naive taildrop causing rFVD degradation. While achieving rFVD comparable to the original baseline, our approach achieves a better gFVD and lower gFVD/rFVD ratio, demonstrating that we effectively reduced the training & inference discrepancy in the AR model through this method.
>
> | Method | Training Technique | Token | rFVD | gFVD | gFVD/rFVD |
> | --- | --- | --- | --- | --- | :---: |
> | LARP | none | 512 | 53.25 | 86.25 | 1.62 |
> | VaporTok | naive taildrop | 512 | 81.94 | 93.34 | 1.14 |
> | VaporTok | probability taildrop | 498 | 53.92 | 80.13 | 1.48 |
>
> ---
>
> ### **Q2-1: Why does naive taildrop still use 1,024 tokens during inference?**
>
> In Appendix C.4, we provide a detailed description of the training and inference procedures for naive taildrop. Specifically：
>
> During training, naive taildrop samples a truncation index uniformly at random, discards all subsequent tokens, and feeds the preceding tokens into the decoder for reconstruction.
>
> During inference, one may apply certain heuristics to determine how many tokens to use for reconstruction (e.g., the target reconstruction threshold employed by ElasticTok), or simply reconstruct using all tokens. Because reconstruction with the full token set always yields better performance than using only a subset, the results we report for naïve taildrop on the evaluation set are obtained by reconstructing with all 1,024 tokens.
>
> ---
>
> ### **Q2-2: Why does naive taildrop cause a degradation in rFVD performance?**
>
> Leveraging taildrop’s semantic-to-detail latent-space properties comes with the drawback that naive taildrop reduces rFVD. For example, the comparison between TiTok[1] and One-D-Piece[2] (TiTok trained with naive taildrop), as shown in Table 2 and Figure 4 of the One-D-Piece paper, demonstrates that using the same number of tokens leads to a drop in rFVD.
>
> - TiTok-L with 32 tokens: rFVD increases from 2.21 (TiTok) to 3.23 (One-D-Piece)
> - TiTok-B with 64 tokens: rFVD increases from 1.71 (TiTok) to 2.39 (One-D-Piece)
> - TiTok-S with 128 tokens: rFVD increases from 1.70 (TiTok) to 1.96 (One-D-Piece)
>
> As for the causes of this reconstruction performance degradation, they may include the following:
>
> - One-D-Piece attributes this phenomenon to the fact that too few tokens cannot deliver good perceptual quality.  Likewise, applying naive taildrop to compress the full set of 1,024 tokens when representing a 16-frame video may similarly undermine its representational capacity.  Our probability taildrop, by contrast, compensates for the rFVD degradation caused by taildrop by adjusting the number of tokens according to the visual complexity of the data, thereby alleviating this issue.
> - Moreover, current methods apply taildrop on the discrete latent representations—i.e., they perform VQ, immediately follow it with taildrop, and then reconstruct via the decoder. This incurs two successive levels of information loss (the per-token quantization loss from VQ and the sequence-level information loss from taildrop), making it more difficult to optimize the VQ codebook. We leave this aspect for future exploration.
>
> [1] An Image is Worth 32 Tokens for Reconstruction and Generation
>
> [2] One-d-piece: Image tokenizer meets quality-controllable compression.
>
> ---
>
> ### **Q3-1: Does the taildrop probability after GRPO  align with that after first stage only optimized by data prior ?**
>
> During the GRPO optimization process, a KL divergence loss term is included to ensure similarity. The KL divergence between the old and new taildrop probability in GRPO loss function ensures that the multi-task-aware taildrop probability remains anchored close to data-prior taildrop probability learned in the first stage.
>
> To verify the above statement, we compute the similarity between the taildrop probability before and after optimized by GRPO over all samples in the UCF101 validation set using the following definition (higher score indicates that the distributions are more similar)：
>
> \begin{align}
> S_m &= \max\bigl(0,100 - \min\bigl(8 |m_a - m_b|\,100\bigr)\bigr)  ,\quad\text{where} \quad m_a= \mathrm{median}(a), m_b= \mathrm{median}(b),\\
> \end{align}
>
> \begin{align}
> S_d &= \bigl(\min\bigl(100\,100 r_d\bigr)\bigr)^{1.2} ,\quad\text{where} \quad \mathrm{IQR}_a= Q_3(a) - Q_1(a), \mathrm{IQR}_b= Q_3(b) - Q_1(b), r_d= \frac{\min(\mathrm{IQR}_a,\mathrm{IQR}_b)}{\max(\mathrm{IQR}_a,\mathrm{IQR}_b)}
> \end{align}
>
> \begin{align}
> S_{\mathrm{KS}} &= \max\bigl(0\,100 - \min\bigl(150 D, 100\bigr)\bigr) ,\quad\text{where} \quad D = \sup_x\lvert F_a(x) - F_b(x)\rvert
> \end{align}
>
> \begin{align}
> S_p &= \max\bigl(0\,100 - \min\bigl(30 |p_a - p_b| \,100\bigr)\bigr) ,\quad\text{where} \quad p_a= \mathrm{peak}(a),  p_b= \mathrm{peak}(b).
> \end{align}
>
> \begin{align}
> Similarity Score &= \tfrac{1}{4}\bigl(S_{m} + S_{d} + S_{\mathrm{KS}} + S_{p}\bigr).
> \end{align}
>
> The results, shown in the table below, indicate that for the vast majority of samples the two distributions exhibit very high similarity. Samples with lower similarity reflect cases where GRPO has performed more extensive exploration in order to reconcile the four distinct reward components. Perceptually, the smooth Gaussian distribution learned in the first stage undergoes peak attenuation and a shift of its mean toward fewer tokens—thereby increasing the opportunity for more tokens to be sampled and becoming more efficient—which is consistent with GRPO’s exploration&exploitation tradeoff.
>
> | Similarity Score  | Counts |
> | :---: | --- |
> | 90-100 | 644 |
> | 80-90 | 384 |
> | 70-80 | 518 |
> | 60-70 | 581 |
> | 50-60 | 593 |
> | 40-50 | 1063 |
> | 30-40 | 0 |
> | 20-30 | 0 |
> | 10-20 | 0 |
> | 0-10 | 0 |
>
> ---
>
> ### **Q3-2: Does the taildrop probability after GRPO reflect data simplicity or complexity?**
>
> The taildrop probabilities after GRPO not only align with those learned in the first stage, but also genuinely reflect the complexity of the data. We apologize that, due to this year’s NeurIPS policy, we are unable to upload visualization figures; however, we can explain this from two perspectives:
>
> From a quantitative perspective, the taildrop probabilities learned in the first stage fully capture the data’s complexity and the GRPO-refined taildrop probabilities are highly similar to those initial values, so the GRPO-refined probabilities likewise reflect the complexity of the video data.
>
> From a qualitative perspective, for example, in a violin-playing video where primarily only the hands move, the mean taildrop probability is around 200 tokens; by contrast, in a soccer video with complex backgrounds and large-scale motion, the mean rises to approximately 700 tokens.
>
> ---
>
> ### **Q3-3: Whether cold-starting the module using the current visual prior is both expected and necessary.**
>
> As noted in our responses to Q3-1 and Q3-2, this data prior serves the role we "expected". As for "necessary", we provide the following explanation:
>
> - Just as large language models undergo pretraining and supervised fine-tuning before GRPO to establish a strong foundation, we also require a well-initialized distribution to GRPO. This initialization stabilizes GRPO’s exploration and accelerates convergence.
> - Since our reward does not include any data-specific term, GRPO alone only trades off efficiency against reconstruction & generation. Without data-prior constraint in the first stage, the final adaptive distribution would not correlate strongly with the underlying data complexity.
>
> In summary, we want our tokenizer to be both data-prior-aware and task-metrics-aware, so we inject the data prior in the first stage. Then, in the second stage, the KL divergence term in the GRPO loss guarantees that while the distribution is adjusted toward task objectives, it retains the information encoded by the data prior. These two phases thus complement one another to achieve our overall goal.

---

> > ### Comment · Reviewer_FNXp · 2025-08-07
> >
> > Thank authors for the detailed rebuttal. I'm satisfied with the response and keep my original rating.

---

> > > ### Author Response · Authors · 2025-08-08
> > >
> > > We sincerely thank the reviewer for the time and thoughtful feedback.

---

### Official Review · Reviewer_R7pP · 2025-07-19

**Clarity:** 4
**Significance:** 4
**Originality:** 3
**Rating:** 5
**Confidence:** 4

**Summary:**

This paper develops VaporTok, a model for adaptive video tokenization that accounts for both content complexity and for downstream tasks like generation. Regarding content complexity, instead of using uniform taildrop (nested dropout), VaporTok learns to predict the number of tokens to use for a video clip based on the content and a visual complexity prior. This information is used during training to better optimize VaporTok to use more tokens for complex scenes and fewer tokens for simple ones.

Regarding awareness of downstream tasks, the authors develop a method for training the taildrop probability query module using GRPO. Learning is driven by four reward signals that capture efficiency (fewer tokens is better), diversity (avoid mode collapse), reconstruction (preserve the autoencoder functionality of the tokenizer), and generation (awareness of the downstream video generation task).

The authors evaluate VaporTok using the UCF-101 and Kinetics-600 video datasets. They show reasonable gFVD results using 50-65% fewer tokens than previous methods (Table 1). Extensive ablation studies are also performed to understand the impact of different design choices (taildrop vs. probabilistic taildrop, sampling method, etc.).

**Questions:**

Q1. How does VaporTok compare to existing methods when using (roughly) the same number of tokens on average? The expectation is that an adaptive method should perform much better than a fixed method on a diverse dataset since it can more efficiently allocate tokens: fewer for simple videos and more for complex videos leading to a net improvement in reconstruction quality.

Perhaps this is shown in Table 2 comparing LARP with 512 tokens to VaporTok with 498 where rFVD is nearly the same and gFVD is slightly betterfor VaporTok? If you agree this is a fair comparison, why isn't there a larger benefit from the adaptive tokenization of VaporTok?

To better motivate my question, adaptive tokenization is a kind of variable rate compression, and the standard visualization for codecs is a rate-distortion curve. For tokenization, the key graph would show FVD vs. average number of tokens. Importantly, it's very difficult to compare different methods from a single point (one rate and one distortion) if one method doesn't "dominate" the other (lower rate and lower distortion). Table 1 doesn't show this since VaporTok has fewer tokens but also worse FVD so it's hard to draw a strong conclusion.


Q2. Does VaporTok support different token counts for a single video? I think it does since the taildrop index is sampled during training so some flexibility should exist at inference time. Assuming that's right, I'd expect to see a visualization that shows how reconstructions change as you move from very few to a relatively large number of tokens for a particular video.

Such a visualization would also make it clear whether early tokens truly encode *semantic* information or if the representation is better described as coarse-to-fine. For example, if you reconstruct a video of a dog with very few tokens, do you get a sharp video of a different dog or a blurry/distorted video of the same dog?

**Ethical Concerns:**

["NO or VERY MINOR ethics concerns only"]

**Final Justification:**

Based on the authors' responses to the initial reviews, I increased my rating from "borderline accept" to "accept".

I think this is justified because adaptive tokenization is a growing research area for generative image and video models, and VaporTok provides a solution to the problem of visual quality degradation due to taildrop, which is a common method for achieving adaptive representations. This approach will be interesting to many researchers in the NeurIPS community.

I don't think that a "strong accept" is justified due to some shortcomings in the current method:

1) The semantic-to-detail hierarchy does not appear to extend to the case where very few tokens are used (contrast with Flex
Tok)

2) While the "probability traildrop" method presented in the paper compensates for the quality degradation caused by "naive taildrop", it's reasonable to believe that an adaptive method will outperform fixed-rate methods (even at moderate-to-high token counts), which is not achieved by VaporTok. So I think probability taildrop is an interesting and important step forward, but there's room for much better solutions to high-quality adaptive tokenization.

**Limitations:**

yes

**Quality:**

3

**Strengths And Weaknesses:**

Quality -- The quality of this paper is generally very high. The methods used make sense, and I see no technical issues with the formulation. The evaluation is quite thorough in terms of ablations, comparing to previous methods, and showing the impact of different reward signals. However, there's a gap in the evaluation related to adaptivity since results are only shown for a single average token count (see the Questions section for more detail).

Clarity -- The paper is very clearly written and well-organized. The authors also provide code.

Significance -- The addresses two problems of interest within video tokenization and generation, and so I think the significance is quite high. First, adaptive tokenization is a very active area of research right now since the memory and computational costs of video modeling (for understanding and generation tasks) is quite high in part because most methods use highly redundant, and thus wasteful, representations. Second, task-aware and joint optimization of multiple components within a generative system is also an active area of research since training components in isolation is unlikely to yield optimal results.

Originality -- Both of the primary contributions of this paper (probabilistic taildrop and RL for task-aware tokenizer training) are novel as far as I know.

---

> ### Author Rebuttal · Authors · 2025-07-30
>
> We thank the reviewer for their recognition of our work and for the constructive comments. We provide our feedback as follows:
>
> ---
>
> ### **Q1-1: How does VaporTok have advantages over other methods?**
>
> Our probabilistic taildrop was configured with a maximum token length of 1024. However, since we utilize adaptive token count during both training and inference, the actual average number of tokens used is less than 1024, averaging around 500 tokens. To enable a fairer comparison, we primarily compare our approach against the 512-token scheme in this context, as follows:
>
> | Method | Training Technique | Token | rFVD | gFVD | gFVD/rFVD |
> | --- | --- | --- | --- | --- | :---: |
> | LARP | none | 512 | 53.25 | 86.25 | 1.62 |
> | VaporTok | naive taildrop | 512 | 81.94 | 93.34 | 1.14 |
> | VaporTok | probability taildrop | 498 | 53.92 | 80.13 | 1.48 |
>
> Our approach offsets the rFVD degradation caused by taildrop, achieving rFVD on par with the baseline. Furthermore, our probabilistic taildrop also exhibits the semantic-to-detail property, so it delivers superior generation performance and lower gFVD/rFVD than the baseline, demonstrating that we effectively mitigate the training-inference gap of the downstream AR generation model.
>
> ---
>
> ### **Q1-2: Why isn't there a larger benefit from the adaptive tokenization of VaporTok?**
>
> When training a visual tokenizer, both reconstruction and generation metrics are important; however, since our goal is to solve the AR generation problem, we place greater emphasis on generation. Taildrop is inherently unfriendly to reconstruction (as shown by the TiTok vs. One-D-Piece reconstruction results below), but by ordering the latent space from semantic to detail it makes the VAE decoder more robust to later tokens, thereby boosting generation. Our method retains this strong generation performance while cleverly overcoming taildrop’s reconstruction weakness, achieving rFVD results on par with the baseline.
>
> This degradation in rFVD performance actually occurs in all naive taildrop methods. For example, the comparison between TiTok[1] and One-D-Piece[2] (TiTok trained with naive taildrop), as shown in Table 2 and Figure 4 of the One-D-Piece paper, demonstrates that using the same number of tokens leads to a drop in rFVD, specifically:
>
> - TiTok-L with 32 tokens: rFVD increases from 2.21 (TiTok) to 3.23 (One-D-Piece)
> - TiTok-B with 64 tokens: rFVD increases from 1.71 (TiTok) to 2.39 (One-D-Piece)
> - TiTok-S with 128 tokens: rFVD increases from 1.70 (TiTok) to 1.96 (One-D-Piece)
>
> As for the causes of this reconstruction performance degradation, they may include the following:
>
> - One-D-Piece attributes this phenomenon to the fact that too few tokens cannot deliver good perceptual quality.  Likewise, applying naive taildrop to compress the full set of 1,024 tokens when representing a 16-frame video may similarly undermine its representational capacity.
> - Moreover, current methods apply taildrop on the discrete latent representations, i.e., they perform VQ, immediately follow it with taildrop, and then reconstruct via the decoder. This incurs two successive levels of information loss (the per-token quantization loss from VQ and the sequence-level information loss from taildrop), making it more difficult to optimize the VQ codebook, thereby leading to decreased rFVD performance. We leave this aspect for future exploration.
>
> Our probabilistic taildrop, by contrast, compensates for the rFVD degradation caused by taildrop by adjusting the number of tokens according to the visual complexity of the data, thereby alleviating this issue. Furthermore, our training approach arranges the video latent space in a semantic-to-detail order and strengthens the decoder’s robustness to later tokens, thereby enhancing generation performance.
>
> [1] An Image is Worth 32 Tokens for Reconstruction and Generation
>
> [2] One-D-Piece: Image Tokenizer Meets Quality-Controllable Compression
>
> ---
>
> ### **Q2: Does VaporTok’s latent space truly exhibit the semantic-to-detail (or coarse-to-fine) property?**
>
> VaporTok supports reconstruction with varying token counts. (For details on the distinction between probabilistic taildrop during training and inference, see Appendix C.4. In brief, during training with probabilistic taildrop we could employ three different modes to get truncation index—argmax, sample, and pre-sample—whereas at inference, to eliminate randomness, we report results obtained exclusively via argmax sampling.) Also, **by directly setting the truncation index, we can control how many tokens are used for reconstruction**. However, if using a fixed token count for inference, VaporTok will lose the token-efficiency advantages provided by its adaptivity.
>
> Due to this year’s NeurIPS policy, we cannot submit a PDF to display the semantic-to-detail visualization. Instead, we implicitly demonstrate this trend through reconstruction metrics at different token counts. In addition to rFVD, we report MSE, PSNR, and LPIPS of VaporTok and LARP(512) at different token counts. **With an extremely small number of tokens, VaporTok significantly outperforms LARP(512), demonstrating that its head tokens capture strong semantic information and exhibits the semantic-to-detail characteristic.**
>
> The reconstruction results of V(VaporTok) and L(LARP-512)：
>
> | Token counts | V-MSE | L-MSE | V-PSNR | L-PSNR | V-rFVD | L-rFVD | V-LPIPS | L-LPIPS |
> | --- | --- | --- | --- | --- | --- | --- | --- | --- |
> | 16 | 0.0400 | 0.1964 | 14.62 | 7.46 | 1774 | 4020 | 0.484 | 0.789 |
> | 32 | 0.0340 | 0.1431 | 15.36 | 8.94 | 1449 | 3816 | 0.441 | 0.775 |
> | 64 | 0.0210 | 0.1603 | 17.51 | 8.34 | 858 | 3825 | 0.346 | 0.765 |
> | 128 | 0.0113 | 0.1479 | 20.24 | 8.68 | 387 | 3222 | 0.240 | 0.661 |
> | 256 | 0.0058 | 0.0235 | 23.35 | 17.53 | 101 | 521 | 0.155 | 0.325 |
> | 512 | 0.0036 | 0.0031 | 25.40 | 26.10 | 60 | 53 | 0.115 | 0.105 |
> | 1024 | 0.0024 | \ | 27.19 | \ | 41 | \ | 0.09 | \ |
>
> As for the qualitative visualizations, our final results are nearly indistinguishable from those shown in One-D-Piece (see Figure 1 of One-D-Piece paper). The critical factor in exhibiting the semantic-to-detail characteristic is using varying token lengths during training: naive taildrop chooses lengths entirely at random, whereas we sample lengths from a learned distribution. Although this sampling can somewhat attenuate the semantic-to-detail effect, we still train with different token lengths , so VaporTok naturally displays the same characteristic with One-D-Piece.

---

### Author Response · Authors · 2025-08-09
**General Response**

We thank the reviewers for their constructive feedback. This note (i) recaps the highlighted strengths and (ii) summarizes issues resolved during the rebuttal.

**Recognized strengths:**

1. **Significance.** Our paper addresses a high-impact problem: **adaptive video tokenization** and **joint optimization that is aware of data priors and downstream tasks**. Adaptivity reduces redundant tokens and computation, and joint training aligns the tokenizer with data prior and downstream tasks, avoiding the suboptimality of training components in isolation. (Reviewer R7pP, T8WN)
2. **Novelty.** As recognized by the reviewers, our method is highly original and novel. We propose **Probabilistic Taildrop** to realize adaptive tokenization and reformulate video encoding as a **GRPO-optimizable sequential decision process**, yielding adaptivity that accounts for both data priors and downstream objectives. (Reviewer R7pP, FNXp, 8AKU)
3. **Effectiveness.** Our method significantly reduces the token count while maintaining both reconstruction and generation performance. (Reviewer R7pP, FNXp, T8WN)
4. **Clarity.** The paper is easy to follow, with a clear problem statement, intuitive motivation, and concise formulas. (Reviewer R7pP, FNXp, 8AKU)

**Resolved during rebuttal:**

1. **Stronger validation.** We added comparisons to **more baselines** (Reviewer 8AKU Q1) and an **SFT ablation** isolating the benefit of RL (Reviewer T8WN Q1). Additionally, we conducted a **human study** to further validate the method’s effectiveness (Reviewer T8WN Q3).
2. **Deeper analysis.** We analyzed hyperparameter sensitivity (Reviewer T8WN Q2), the overhead of the additional modules (Reviewer 8AKU Q2-1), the alignment of the learned taildrop probabilities with the data prior (Reviewer FNXp Q3-2), and convergence behavior (Reviewer 8AKU Q2-1). Results show **mild** **sensitivity** within common ranges, **modest** **overhead**, and **stable** **convergence**. Moreover, to clarify points that were not fully understood, we detailed our consistency comparisons against baselines and naive taildrop (Reviewer R7pP Q1-1, Reviewer FNXp Q1) and the implementation details and known drawbacks of naive taildrop (Reviewer R7pP Q1-2, Reviewer FNXp Q2).
3. **Additional quantitative evidence.** New experiments substantiate key claims: (a) the latent space exhibits a **semantic-to-detail** progression (Reviewer R7pP Q2); (b) taildrop probabilities are **similar before and after RL**, indicating consistent truncation behavior (Reviewer FNXp Q3-1).

We appreciate the reviewers’ comments and believe these changes resolve the concerns raised.

---

### Decision · Program_Chairs · 2025-09-17

**Decision:**

Accept (poster)

**Comment:**

The paper proposes VaporTok, an adaptive video tokenizer combining probabilistic taildrop with GRPO to allocate tokens by content complexity and downstream objectives. It received 1 A and 3 BAs. Reviewers largely agree on novelty and practical impact, noting substantial token reductions at comparable reconstruction quality and improved generation metrics at similar average token counts; added analyses (SFT vs GRPO, sensitivity, overhead), a human study, and broader comparisons address earlier concerns about baselines and complexity. One reviewer still points out that the method does not strictly dominate fixed-rate baselines across the full rate–distortion range and that the semantic-to-detail claim is not fully established at very low token counts; however, these are outweighed by the method’s soundness, clarity, code availability, and demonstrated efficiency gains on UCF101/K600. As a result, the AC recommands accept.